# Pursuing Minimal Sufficiency in Spatial Reasoning

**Yejie Guo**[1*†]**, Yunzhong Hou**[2*]**, Wufei Ma**[3]**, Meng Tang**[4]**, Ming-Hsuan Yang**[4]

[1]Shanghai Jiao Tong University   [2]Beijing Institute of Technology
[3]Johns Hopkins University   [4]University of California Merced

## Abstract

Spatial reasoning, the ability to ground language in 3D understanding, remains a persistent challenge for Vision-Language Models (VLMs). We identify two fundamental bottlenecks: *inadequate* 3D understanding capabilities stemming from 2D-centric pre-training, and reasoning failures induced by *redundant* 3D information. To address these, we first construct a Minimal Sufficient Set (MSS) of information before answering a given question: a *compact* selection of 3D perception results from *expert models*. We introduce **MSSR** (Minimal Sufficient Spatial Reasoner), a dual-agent framework that implements this principle. A *Perception Agent* programmatically queries 3D scenes using a versatile perception toolbox to extract sufficient information, including a novel **SOG** (Situated Orientation Grounding) module that robustly extracts language-grounded directions. A *Reasoning Agent* then iteratively refines this information to pursue minimality, pruning redundant details and requesting missing ones in a closed loop until the MSS is curated. Extensive experiments demonstrate that our method, by explicitly pursuing both sufficiency and minimality, significantly improves accuracy and achieves state-of-the-art performance across two challenging benchmarks. Furthermore, our framework produces interpretable reasoning paths, offering a promising source of high-quality training data for future models. Source code is available at https://github.com/gyj155/mssr.

## 1 Introduction

Spatial reasoning—the ability to perceive and reason about object relationships in 3D space—is a cornerstone of general intelligence and a critical prerequisite for deploying AI in the physical world, from robotics to AR/VR (Cheng et al., 2024a; Kim et al., 2024). While modern VLMs (OpenAI, 2024; DeepMind, 2025) have achieved remarkable success, they still consistently fail on spatial reasoning tasks (Yang et al., 2025c; Li et al., 2025a). In this work, we diagnose this critical gap by identifying two fundamental bottlenecks:

**Inadequate 3D perception.** Trained predominantly on 2D data, VLMs lack geometric priors and thus struggle to perceive 3D information like layout, orientation, and depth (Ma et al., 2023; 2024b).

**Redundancy degrades reasoning.** 3D environments are information-dense. Naively aggregating all percepts floods the context with weakly relevant details, potentially diluting attention (Liu et al., 2023) and encouraging shortcut heuristics (Xiao et al., 2024)—ultimately degrading performance (Fig. 1).

In the face of these two challenges, cognitive science offers a compelling insight: humans navigate complex scenes not by exhaustively processing all sensory data, but by constructing task-specific, minimal mental models (Tversky, 1991). Based on the mental models, they then selectively attend to the details needed to make a decision, and incrementally update the mental models as required (Byrne & Johnson-Laird, 1989; Johnson-Laird, 2010). This principle is formalized in statistics by

---

*Equal contribution. †Work done while visiting UC Merced.

**Heuristic Shortcuts**

Q. Is the chair facing the window?

```
Non-minimal Set:
yaw(chair)≈+90°
window position
office desk is in front of chair
notebook and mug on desk
Reasoning:
Desk is in front of chair, as chair
typically faces desk...No. ✖
```

**Attention Dilution**

Q. Is the chair between the desk and the window?

```
Non-minimal Set:
position (chair, desk, window)
chair near to the door
desk on the right
window on the left wall
Reasoning:
Desk is on the right, chair near to the door.
There is no direct line from this...No. ✖
```

Figure 1: Irrelevant (highlighted in red) information can overwhelm the VLM and hurt its performance. **Left**: VLM ignores crucial information and hallucinates. **Right**: VLM cannot attend to the important part.

the Minimal Sufficient Statistic (Lehmann & Casella, 1998), which captures all relevant information from a sample in the most compressed form possible.

We show that the key to robust spatial reasoning lies in a similar pursuit: actively discovering a **Minimal Sufficient Set (MSS)**—the most compact representation of spatial information required to answer a specific query. Motivated by this principle, we introduce **MSSR** (**M**inimal **S**ufficient **S**patial **R**easoner), a zero-shot framework that operationalizes this pursuit through a dual-agent architecture. MSSR disentangles the challenges of perception and reasoning into two specialized, collaborative agents:

To bridge the 3D perception gap, we equip a *Perception Agent* (PA) with a suite of vision modules to programmatically query the scene for spatial primitives (*e.g.*, locations, directions, relations) and return a structured, VLM-friendly state. A key limitation of existing tool-augmented VLMs is their inability to ground complex, situational directions specified by language descriptions. We address this with a **SOG** (**S**ituated **O**rientation **G**rounding) module, which reformulates the orientation estimation task as a multi-choice question, and to overlay candidate 3D directions on 2D images as visual prompting. Through a procedural coarse-to-fine approach and alternative 3D view rendering to eliminate ambiguities, it robustly extracts object orientations ("Is the chair facing the door?") and behavior-centric directions ("Which way is the person facing while ascending the stairs?"). Thus, the PA provides rich and accurate 3D data without costly end-to-end training.

While the PA gathers extensive perceptual data, it risks worsening the second bottleneck: information redundancy that could degrade reasoning accuracy. To solve this, a *Reasoning Agent* (RA) strategically prunes this stream of information and curates the MSS by explicitly filtering redundant or task-irrelevant 3D information. It first formulates a high-level reasoning plan, assesses the collected information, and subtracts non-contributing information. If this set is deemed insufficient to confidently answer the question, the RA issues targeted, specific requests back to the PA to acquire only the missing information. This iterative refinement continues until a MSS is formed. The final answer is derived exclusively from this curated set, sharpening the model's focus and mitigating errors from distracting data.

We evaluate MSSR on two challenging benchmarks: MMSI-Bench (Yang et al., 2025c), which tests situated multi-view reasoning in complex scenes, and ViewSpatial-Bench (Li et al., 2025a), which focuses on perspective relational understanding. MSSR achieves state-of-the-art performance against strong monolithic and agentic baselines, while producing compact, interpretable reasoning paths. Extensive ablation studies demonstrate the harmful effect of redundancy and the efficacy of the RA's pruning in restoring performance. Our contributions are threefold:

- We formulate 3D spatial reasoning as **Minimal Sufficient Set construction** and introduce a **dual-agent** framework that interleaves perception with high-level planning to acquire *just enough* information.
- We design a **perception agent** that has access to a versatile toolbox including a SOG module for robust directional grounding; and a **reasoning agent** that directs the entire process and provides final answers.
- Our MSSR effectively improves the 3D spatial reasoning performance on two challenging benchmarks and produces interpretable reasoning traces that can provide supervision for future 3D-aware models.

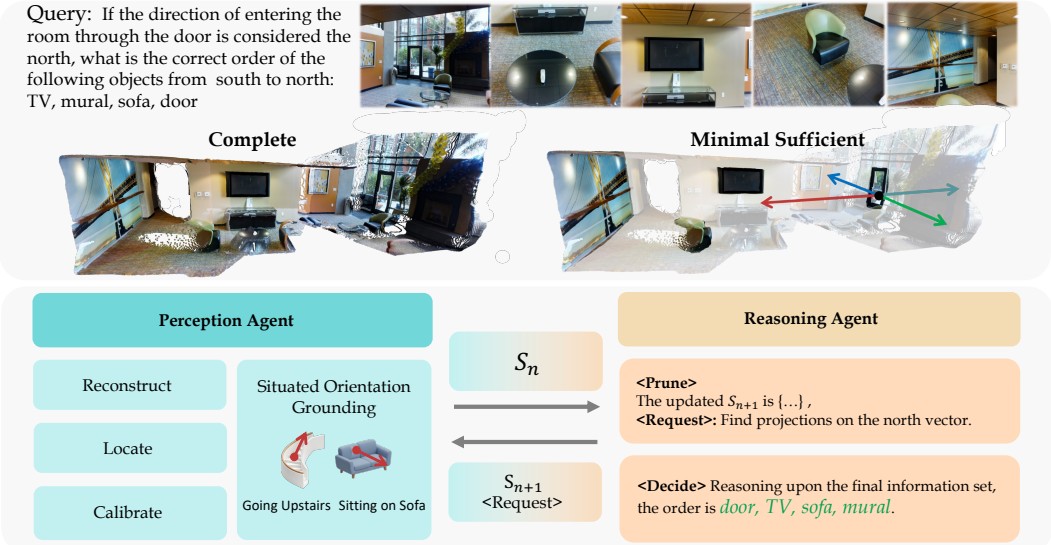

Figure 2: The MSSR dual-agent framework. **Top**: To avoid overwhelming the VLM, MSSR extracts a Minimal Sufficient Set that only contains the necessary information. **Bottom**: Dual-agent loop: the Perception Agent provides a spatial information set ($S_n$), and the Reasoning Agent prunes irrelevant information, requests missing information, or makes a final decision based on the curated set.

## 2 RELATED WORK

**Monolithic VLMs in Spatial Reasoning** typically inject 3D knowledge by fine-tuning on synthetic data (Chen et al., 2024a; Cheng et al., 2024b; Ma et al., 2025a) or integrate specialized modules to process 3D modalities like point clouds (Hong et al., 2023; Huang et al., 2024; Ma et al., 2025b). While demonstrating progress, these methods are fundamentally limited: they require prohibitively expensive 3D instruction datasets (Yang et al., 2025b) and risk forgetting pre-trained knowledge, which degrades the VLM's crucial general-purpose reasoning abilities (Kirkpatrick et al., 2017). In contrast, MSSR is a zero-shot, training-free framework that bypasses these issues. By preserving the VLM's full capabilities and instead explicitly structuring the perception-reasoning process, our approach maximizes spatial reasoning performance without compromising the model's versatility or requiring costly data and retraining.

**Agentic Frameworks** are a prominent line of work where complex problems are decomposed into steps solved via tool-use. Pioneering methods like ReAct (Yao et al., 2023) showed that LLMs can interleave reasoning and action, a paradigm extended to 3D where agents gather spatial information for tasks like embodied exploration (Yang et al., 2025d) and 3D VQA (Ma et al., 2024a). These approaches focus primarily on information gathering, often adopting a purely accumulative strategy. However, for the tasks we address, the dense nature of 3D scenes introduces significant redundancy, where an excess of irrelevant spatial details degrades performance. MSSR is therefore designed to not only gather information, but also prune irrelevance, which is a key departure from prior agentic designs.

**Visual Programming** is a paradigm often used for 3D attribute querying (Marsili et al., 2025; Yuan et al., 2024). It enhances VLMs by decomposing complex visual tasks into executable programs that leverage specialized modules (Gupta & Kembhavi, 2023; Surís et al., 2023). We adopt the visual programming as the execution backbone for our Perception Agent, leveraging its modularity to integrate specialized tools. Instead of the typical one-shot execution, our framework advances this paradigm by integrating visual programming into a closed loop. By preserving the full execution state across iterations, subsequent perception steps can build upon prior computations, enabling dynamic information refinement while avoiding redundant work.

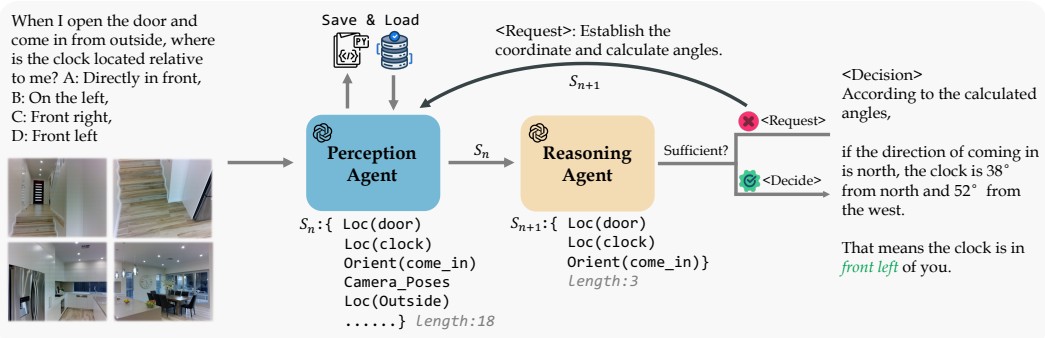

Figure 3: **A detailed example.** The Perception Agent provides an 18-item set $S_n$, and the Reasoning Agent prunes to 3 essential items in $S_{n+1}$. Deeming this insufficient, the RA issues a `<Request>` for missing calculations. With sufficient information, it makes a final `<Decision>`.

## 3 METHOD

We address *language-conditioned spatial reasoning*: given $M$ views $\mathcal{I} = \{I_1, \ldots, I_M\}$ from the same scene and a natural-language query $q$, the goal is to produce the answer $a$. Before answering the 3D reasoning question, our approach first builds a Minimal Sufficient Set (MSS)—a compact representation that is both sufficient to answer the query and minimal to prevent failures from redundant and distracting information.

### 3.1 OVERVIEW

As illustrated in Figure 2, our method actively curates the MSS. The process is iterative, interleaving information acquisition and strategic pruning to converge on a representation that is both sufficient and minimal. Formally, we define the target of this process, the MSS, as follows: Let $\mathcal{W}$ represent the comprehensive set of all spatial and semantic information derivable from the full 3D scene. We consider a set of spatial information $\mathcal{S} \subseteq \mathcal{W}$, which is a subset of $\mathcal{W}$. Our goal is to find the MSS $\mathcal{S}^\star \subseteq \mathcal{W}$, which satisfies two properties:

1. **Sufficiency**: Ideal MSS $\mathcal{S}^\star$ must contain enough information for an oracle reasoning agent, $\mathcal{R}^\star$, to correctly answer the query $q$. Formally, this means:

$$\mathcal{R}^\star(\mathcal{S}^\star, q) = a^\star \tag{1}$$

where $a^\star$ is the ground-truth answer. This ensures that no essential information for reasoning is omitted.

2. **Minimality**: MSS $\mathcal{S}^\star$ should be free of redundant or irrelevant information that would burden the reasoning model. Ideally, it is the smallest such set that maintains sufficiency. This can be expressed as:

$$\forall \mathcal{S}' \subset \mathcal{S}^\star, \quad \mathcal{R}^\star(\mathcal{S}', q) \neq a^\star \tag{2}$$

Through updating the spatial information set $\mathcal{S}$, we aim to approximate the MSS $\mathcal{S}^\star$. In this attempt, MSSR conducts a closed-loop collaboration between the Perception Agent (PA) and Reasoning Agent (RA), as shown in Figure 3. The process begins with an empty $\mathcal{S}$ and proceeds iteratively. Initially, the PA executes a broad perception directive to populate $\mathcal{S}$ with a comprehensive, potentially non-minimal set of spatial primitives. This set is then passed to the RA for curation, where it formulates a reasoning plan and prunes any information not causally linked to the plan, thereby enforcing minimality. If the pruned set is deemed insufficient, the RA performs an assessment and issues a targeted request back to the PA for precisely the missing information, which then augments $\mathcal{S}$. This cycle of curation and targeted augmentation repeats until the RA judges $\mathcal{S}$ to be sufficient for answering the query. At this final stage, the RA discards all prior context and reasons exclusively over the curated MSS to produce the answer, ensuring both focus and interpretability.

## 3.2 PERCEPTION AGENT

The Perception Agent (PA) serves as the perception engine in our framework, responsible for bridging the gap between high-level reasoning directives and raw elements from the 3D scene. To equip the PA with robust and precise 3D perception capabilities, we adopt the **Visual Programming** (Gupta & Kembhavi, 2023) paradigm. The PA is provided with a suite of pre-designed modules, which act as specialized tools. These modules leverage vision expert models for tasks like geometric reconstruction and object localization, significantly augmenting the LLM's capabilities for complex spatial computations. Furthermore, the structured nature of code generation within the visual programming framework ensures that the information extraction process is logical, transparent, and reproducible.

At each turn, the PA receives the current $S$, the original query, scene images $I$ and a natural language request $r$ from the Reasoning Agent representing the current information-gathering goal. Its objective is to generate a Python script that invokes the appropriate foundation modules to fulfill the directive. The process begins with an empty $S$ and an initial, broad instruction, such as:

"`Extract all potentially relevant information to solve the problem. Your goal is to find as much information as possible.`"

The generated script populates a designated dictionary with newly extracted information—such as object coordinates and spatial relationships. This dictionary is then merged into $S$. Crucially, after each execution, the entire state of the Python environment, including all intermediate variables and data structures, is preserved as a **snapshot**. When the PA is invoked in a subsequent turn, this snapshot is reloaded. This mechanism allows the PA to contextually build upon its previous computations, avoiding redundant processing and enabling a more complex, stateful exploration of the scene. The newly updated $S$ is then passed to the Reasoning Agent.

**Spatial Reasoning Modules.** The Perception Agent implements directives by invoking a curated suite of spatial modules, which bridge high-level goals and raw perceptual data to construct the MSS. Following established designs (Surís et al., 2023), our toolkit includes basic modules: a locate module that utilizes vision expert models (Liu et al., 2024; Ravi et al., 2024) to pinpoint object coordinates in 3D, and a computation module that offloads complex numerical tasks, such as coordinate frame transformations. More critically, to address challenges often overlooked by prior work, we propose novel modules designed for more robust spatial reasoning. These advanced modules cover the full pipeline of spatial understanding, from establishing a coherent 3D representation to grounding language-conditioned attributes. The specific implementation of each module is detailed in Appendix C.

**Foundational 3D Scene Reconstruction.** To bridge the critical gap between sparse 2D images and a coherent 3D scene representation, this module leverages recent breakthroughs in rapid neural-based reconstruction models (Wang et al., 2024b;a; 2025) to estimate the camera parameters, depth maps, and a unified 3D point cloud of the scene. In our experiment, VGGT (Wang et al., 2025) shows robust performance and high speed. This output serves as the foundational canvas upon which subsequent spatial information is extracted. Similar to Chen et al. (2024a), we additionally segment the ground plane during reconstruction.

**Global Coordinate System Calibration.** To resolve the inherent ambiguity of view-dependent spatial terms (*e.g.*, "left," "behind"), this module establishes a unified global coordinate system. It aligns the scene's axes based on a reference vector, which is derived either from explicit instructions in the query ("assume the window faces east") or from prominent landmarks. This calibration ensures all directional and relational information stored in the MSS is consistent and unambiguous, a prerequisite for reliable multi-step reasoning.

**Situated Orientation Grounding (SOG).** Beyond simple localization, reasoning often requires understanding orientation. We introduce the SOG module, which utilizes *Visual Prompting* (Qi et al., 2025) to ground complex, language-conditioned directional concepts into 3D vectors. Critically, SOG handles not only intrinsic object orientation ("the front of the chair") but also situation-dependent orientations ("the direction to exit the room")—a capability largely overlooked in prior work. This module dramatically expands the range of addressable queries from static localization to dynamic, perspectival reasoning.

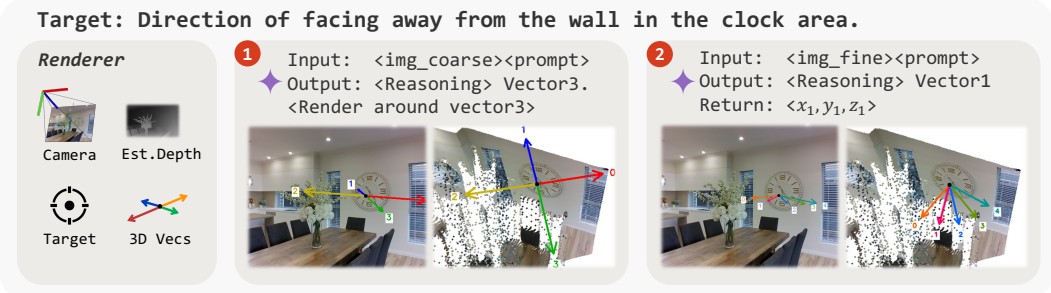

Figure 4: Situated Orientation Grounding as a multi-choice question. We prompt a VLM to select a direction from multiple candidates using the original image and a rendered image, both overlaid with the candidate's 3D orientation vectors. Starting from coarse directions, it then refines this direction to a fine-grained result.

The power of SOG lies in bridging the gap between a VLM's rich semantic scene understanding and its inability to directly regress 3D geometric outputs. Instead of attempting this intractable regression, we reframe orientation grounding as a more reliable *visual selection task*, implemented through a coarse-to-fine strategy as illustrated in Figure 4.

For a query anchored at object position $P_o$, we first randomly generate a sparse set of four coplanar, orthogonal vectors $\{\vec{d_i}\}_{i=1}^4$ parallel to the ground plane, resembling compass directions. To provide the VLM with sufficient context and resolve perspective ambiguity, we render these vectors onto two distinct views: a **Situated View** using the original image to preserve natural context, and a synthetic **Canonical View** from an elevated perspective to reduce foreshortening. The VLM is prompted to select the candidate that best aligns with the language query. This selection is subsequently refined by generating a denser set of candidates around the chosen vector and repeating the selection process, allowing the system to converge on a precise direction. Empirically, this coarse-to-fine strategy robustly and efficiently identifies the target 3D orientation. While not designed for sub-degree accuracy, our experiments confirm that this level of precision is sufficient for most challenging spatial tasks. The vector is then added to $\mathcal{S}$, empowering the RA to tackle a wide range of orientation-dependent problems that were previously intractable.

## 3.3 REASONING AGENT

The Reasoning Agent (RA) acts as the cognitive core in MSSR, responsible for ensuring the information set $\mathcal{S}$ is both sufficient and minimal. At each step, the RA receives the current information set $\mathcal{S}_n$ and the original query $q$. It operates in a two-stage process of information curation and strategic decision-making.

In the first stage, *Plan-Guided Information Curation*, the RA formulates a high-level reasoning plan outlining the necessary steps to answer the query. With this implicit plan, it initializes an empty, updated information set, $\mathcal{S}_{n+1}$. The RA is then prompted to systematically scrutinize each item within the current set $\mathcal{S}_n$, critically evaluating its relevance to the reasoning plan. Only those deemed necessary for the plan are preserved and added to $\mathcal{S}_{n+1}$. This subtractive filtering step is crucial for maintaining the conciseness of $\mathcal{S}$, aggressively pruning any information that is irrelevant to the specific query. After completing the curation, the RA enters the second stage, *Strategic Decision-Making*. Here, it makes one of two decisions:

<Request>: If the RA determines that $\mathcal{S}_{n+1}$ is insufficient to complete the reasoning plan, it then formulates a targeted, natural-language directive that precisely articulates the missing information (*e.g.*, '<Request> The facing direction of someone sitting on the chair.'). This request, along with the pruned $\mathcal{S}_{n+1}$, is passed back to the Perception Agent. The PA uses this focused guidance and updated set to initiate a new round of programming, generating $\mathcal{S}_{n+2}$ that populates the requested information.

<Decide>: Conversely, if the RA concludes that $\mathcal{S}_{n+1}$ contains all necessary information to derive a final answer, it triggers the <Decide> action. It then discards all prior context and reasons exclusively over this final, minimal set using Chain-of-Thought (CoT) (Wei et al., 2022) to produce

Table 1: **Comparison with baselines on MMSI-Bench and ViewSpatial-Bench.** Our method performs favorably against previous methods on these challenging spatial reasoning tasks. Blue is to highlight the improvement of our method over GPT-4o backbone.

| | MMSI-Bench | | | | ViewSpatial-Bench | | |
|---|---|---|---|---|---|---|---|
| | positional relationship | multi-step reasoning | attribute & motion | overall | camera based | person based | overall |
| *Proprietary LLM* | | | | | | | |
| o3 | 45.8 | 34.9 | 36.4 | **41.0** | 51.3 | 51.0 | **51.1** |
| Gemini 2.5 Pro | 38.5 | 34.3 | 36.1 | 37.0 | 44.3 | 41.6 | 43.0 |
| Gemini 2.5 Flash | 37.4 | 30.3 | 33.9 | 35.0 | 41.6 | 36.9 | 38.4 |
| GPT-4o *(backbone)* | 28.0 | 30.8 | 34.3 | 30.3 | 33.6 | 36.3 | 35.0 |
| *Open-source LLM* | | | | | | | |
| Llama-3.2-11B-Vision | 26.9 | 19.2 | 23.9 | 24.5 | 23.7 | 33.6 | 28.8 |
| LLaVA-OneVision-7B | 28.0 | 11.6 | 27.1 | 24.5 | 28.5 | 26.6 | 27.5 |
| Qwen2.5-VL-3B | 29.5 | 23.2 | 23.2 | 26.5 | 39.8 | 32.1 | 35.8 |
| Qwen2.5-VL-7B | 25.9 | 25.8 | 26.1 | 25.9 | 40.6 | 33.4 | 36.9 |
| Qwen2.5-VL-72B | 31.2 | 27.3 | 32.1 | 30.7 | 47.6 | 38.9 | 43.1 |
| Qwen3-VL-4B | 32.8 | 27.8 | 30.3 | **31.1** | 42.3 | 35.9 | 39.0 |
| Qwen3-VL-8B | 32.6 | 30.3 | 28.9 | **31.1** | 45.1 | 37.5 | 41.2 |
| InternVL2.5-8B | 29.9 | 30.3 | 25.4 | 28.7 | 46.5 | 40.2 | **43.2** |
| InternVL3-14B | 25.5 | 29.3 | 27.5 | 26.8 | 47.1 | 33.9 | 40.3 |
| *3D-VLM* | | | | | | | |
| Video-3D-LLM | 25.5 | 25.8 | 23.9 | 25.3 | 24.4 | 35.8 | 30.3 |
| LLaVA-3D | 29.5 | 19.7 | 31.0 | 28.0 | 39.6 | 28.3 | 33.8 |
| VLM-3R | 33.0 | 30.3 | 31.4 | **32.0** | 43.8 | 34.9 | **39.2** |
| *Specialist* | | | | | | | |
| LEO | 42.3 | 32.3 | 38.6 | **39.3** | 41.5 | 45.8 | **43.7** |
| *Agentic* | | | | | | | |
| ViLaSR | - | - | - | 30.2 | 42.4 | 34.4 | 38.2 |
| VADAR | 32.8 | 22.7 | 26.1 | 28.9 | 34.2 | 33.2 | 33.7 |
| *MSSR (Ours)* | 50.6 | 50.0 | 47.1 | **49.5 (+19.2)** | 51.0 | 54.4 | **51.8 (+16.8)** |

the answer. This disciplined use of only the pruned set ensures that the final reasoning is efficient and shielded from the distracting influence of irrelevant data.

Notably, unlike many current 3D agentic systems (Li et al., 2025b; Marsili et al., 2025), both our agents operate in a zero-shot fashion. They are guided by high-level principles rather than in-context learning (ICL) examples. This design endows our method with strong generalization capabilities and mitigates the risk of overfitting to dataset-specific exemplars.

## 4 EXPERIMENTS

### 4.1 BENCHMARKS AND BASELINES

We evaluate our method and various baseline models on two challenging spatial reasoning benchmarks. **MMSI-Bench** (Yang et al., 2025c) is a hand-crafted multi-image dataset focusing on spatial reasoning. The tasks are designed to probe a model's understanding of the positions, attributes, and motions of three elements—cameras, objects, and regions—within real-world environments. Furthermore, it incorporates a multi-step reasoning split, which composes elementary tasks into long-horizon questions to test for deeper spatial reasoning. **ViewSpatial-Bench** (Li et al., 2025a) complements this by addressing a critical limitation of the gap between egocentric and allocentric spatial reasoning. Designed to evaluate multi-viewpoint spatial localization and recognition capability, this benchmark is structured into five distinct task categories to rigorously assess a model's ability to generalize across different spatial viewpoints. These two benchmarks provide a robust, multifaceted platform for testing spatial reasoning capabilities that we aim to advance.

To provide a comprehensive evaluation, we establish four distinct categories of baseline models for comparison: *Proprietary LLMs:* We include powerful, closed-source models with strong reasoning capabilities, such as GPT-4o (OpenAI, 2024), Gemini-2.5-flash (DeepMind, 2025), Gemini-2.5-pro (DeepMind, 2025), and o3 (OpenAI, 2025). *Open-Source LLMs:* We compare with state-of-the-art models such as LLaMA-3.2-Vision (Meta, 2024), LLaVA-OneVision (Li et al., 2024), Qwen2.5-VL (Bai et al., 2025), Qwen3-VL (Yang et al., 2025a), InternVL3 (Zhu et al., 2025), InternVL2.5 (Chen et al., 2024b) and DeepSeek-VL2 (Wu et al., 2024). *Specialists:* We compare with LEO (Huang et al., 2024), a multi-modal generalist model recognized for its proficiency in various 3D tasks. *3D-VLMs:* We evaluate recent 3D-VLMs including VLM-3R (Fan et al., 2025),

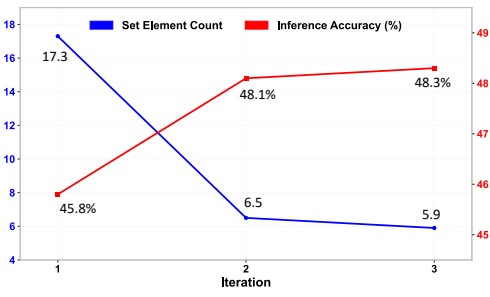

Figure 5: Effects of conciseness on accuracy.

Table 2: **Ablation study of our framework's key components.** We report accuracy (%) on MSR sub-task, MMSI-Bench, ViewSpatial-Bench .

| Method | MSR (MMSI-Bench) | MMSI Bench | ViewSpatial Bench |
|---|---|---|---|
| GPT-4o | 30.8 | 30.3 | 35.0 |
| Ours (Full) | **50.0** | **49.5** | **51.8** |
| Only PA | 33.8 | 37.1 | 32.5 |
| Only RA | 31.8 | 31.1 | 35.3 |
| w/o SOG | 47.0 | 46.9 | 43.2 |
| w/o Iteration | 44.4 | 47.2 | 48.8 |

Video-3D-LLM (Zheng et al., 2025), and LLaVA-3D (Zhu et al., 2024), which are trained and infer with augmented 3D feature representations, making them directly relevant to spatial tasks. *Agents:* We consider VADAR (Marsili et al., 2025), a visual programming framework targeted at spatial inference, and ViLaSR (Wu et al., 2025), a model notable for its ability of visual manipulation.

## 4.2 MAIN RESULTS

On the MMSI-Bench, our method achieves an overall accuracy of **49.5%**. We outperform even the strongest LLM evaluated, o3 (41.0%), demonstrating an absolute gain of 8.5 percentage points.

Compared to the best-performing open-source LLM, Qwen3-VL-8B (31.1%), our approach shows a pronounced relative improvement of over 60%. When contrasted with state-of-the-art 3D-VLM models such as VLM-3R (32.0%), generalist model such as LEO (39.3%), and agentic frameworks such as ViLaSR (30.2%), our method maintains a clear lead.

Transitioning to the ViewSpatial-Bench, which focuses on multi-viewpoint spatial localization and recognition, our method again sets a new benchmark with an overall accuracy of **51.8%**. The consistent strength observed in both Camera Based (51.0%) and Person Based (54.4%) tasks highlights our method's robust generalization across varied spatial viewpoints. Excelling in both categories indicates MSSR's proficiency in bridging the gap between egocentric and allocentric spatial understanding, a limitation for current methods.

## 4.3 ABLATION STUDY

**Effects of Minimality.** A central idea of our framework is that pursuing minimality is crucial for robust reasoning. To rigorously test this, we conducted a controlled ablation study on a representative subset of MMSI-Bench problems MSSR solved in three iterations. For these problems, we created sufficiency-normalized information sets for each step by retrospectively adding critical information (discovered in iterations) to earlier, larger sets. This isolates the effect of set size on the RA's performance, ensuring each set contains the same level of sufficiency. The RA was then tasked to solve the problem independently using these three sets, corresponding to the state after each iteration.

As illustrated in Figure 5, the results reveal a clear inverse correlation between information set size and accuracy. As our iterative pruning strategy reduces the average set element count from an initial 17.3 to a concise 5.9, the RA's inference accuracy concurrently rises from 45.8% to 48.3%. This finding provides empirical evidence that excess information is a significant distractor for LLM-based agents. This affirms that pursuing minimality is fundamental for high-fidelity spatial reasoning, not merely an efficiency optimization.

**Component Analysis.** We conduct ablations to quantify the contributions of each component. In Table 2, we report accuracy on the representative Multi-step Reasoning sub-task and the overall score on MMSI-Bench and ViewSpatial-Bench.

**Only PA:** Removing the RA and tasking the PA to programmatically deduce the answer after information extraction leads to a significant performance drop. In fact, its sequential, top-down

Table 3: **Performance across different LLM backbones.**

| RA Backbone | PA Backbone | MSR (MMSI-Bench) | MMSI Bench | ViewSpatial Bench |
|---|---|---|---|---|
| *Uniform* | | | | |
| LLaVA-OneVision-7B | LLaVA-OneVision-7B | 33.6 | 35.0 | 34.1 |
| Qwen2.5-VL-7B | Qwen2.5-VL-7B | 38.6 | 39.8 | 40.2 |
| Qwen3-VL-8B | Qwen3-VL-8B | 44.2 | 43.1 | 46.5 |
| GPT-4o | GPT-4o | **50.0** | **49.5** | **51.8** |
| *Cross-Model* | | | | |
| GPT-4o | Qwen2.5-VL-7B | 41.3 | 40.1 | 43.0 |
| Qwen2.5-VL-7B | GPT-4o | 44.4 | 44.2 | 46.6 |

execution flow, while effective for information gathering, is less effective in reasoning and question answering, as it often returns sub-optimal results.

**Only RA:** Conversely, removing the PA and forcing the RA to rely solely on the initial context yields negligible improvement over the baseline. This result affirms that prompting alone cannot substitute for the precise, targeted 3D scene perception provided by the PA, even when using a powerful reasoning model. This highlights the crucial synergy between the two agents.

**w/o SOG:** We replaced SOG with a baseline directly prompting the VLM to infer directional vectors, causing an overall performance drop to 46.9%. Our observations suggest this degradation stems from the VLM's inability to regress 3D coordinates, rather than a failure in semantic understanding. This result validates our design of SOG, which circumvents this limitation by reframing regression as a visual selection task.

**w/o Iteration:** Setting the maximum iteration count to 1 forces the RA to make an immediate decision without the ability to request additional information. This leads to a noticeable drop in performance. The degradation is more pronounced on the MSR sub-task, underscoring that for intricate multi-step tasks, the iterative feedback loop is especially vital for achieving information sufficiency.

## 4.4 GENERALIZABILITY ACROSS LLM BACKBONES

To validate the generalizability and scalability of MSSR beyond proprietary models, we evaluated our framework using state-of-the-art open-source backbones, including LLaVA-OneVision-7B (Li et al., 2024), Qwen2.5-VL-7B (Bai et al., 2025), and Qwen3-VL-8B (Yang et al., 2025a). We explored both *uniform* configurations (same backbone for PA and RA) and *cross-model* configurations to analyze agent-specific sensitivity in table 3.

**Consistent Framework Gains.** MSSR consistently boosts performance regardless of the backbone. For instance, while the Qwen2.5-VL-7B baseline achieves 25.9% on MMSI-Bench (see Table 1), integrating it into MSSR improves accuracy to 39.8% (**+13.9%**). The performance scales predictably with model capability (LLaVA-OneVision-7B < Qwen2.5-VL-7B < Qwen3-VL-8B < GPT-4o), demonstrating that MSSR's performance scales consistently with the underlying backbone's strength. This validates the framework's scalability and suggests that future improvements in language models can directly translate to stronger spatial reasoning performance within MSSR.

**Agent Sensitivity.** Cross-model ablations indicate that the Perception Agent is more sensitive to backbone capability than the Reasoning Agent. Downgrading the PA (GPT-4o → Qwen2.5) causes a larger drop (-9.4%) than downgrading the RA (-5.3%). This suggests that the PA's requirement for precise code generation and API utilization demands stronger model capabilities, whereas the RA's natural language planning is relatively more robust.

**Cost-Effective Deployment.** This asymmetry enables a strategic trade-off: using a strong model for the PA and a lighter model for the RA (e.g., GPT-4o + Qwen2.5) retains 90% of the performance (44.2% vs 49.5%) while significantly reducing inference costs.

Table 4: **Accuracy on MMSI-Bench after fine-tuning with data annotated by our framework.**

| Model | Accuracy (%) |
|---|---|
| Qwen2.5-VL-7B | 25.9 |
| Qwen2.5-VL-7B-SFT (annotated by MSSR) | **30.1** (+4.2) |

## 4.5 APPLICATION: ANNOTATING SPATIAL REASONING DATA

Beyond zero-shot inference, **MSSR** serves as a powerful data annotation engine. In addition to numerical results (Hu et al., 2024), its final output—a Minimal Sufficient Set (MSS) and an explicit reasoning trace—provides a rich foundation for creating CoT (Wei et al., 2022)-style data.

To demonstrate this potential, we curated a dataset by sampling 300 (30%) correctly solved questions from our MMSI-Bench runs. We then employed GPT-4o for automated quality filtering, which scrutinized the RA's reasoning for logical consistency and removed cases of incidental correctness, yielding 258 high-fidelity traces. GPT-4o then synthesized these traces into CoT-style annotations. This process grounds the RA's abstract reasoning in gathered spatial information by systematically interleaving its high-level strategic steps (*e.g*., "I need to determine the chair's position") with the specific perceptual evidence from the MSS that substantiates each step (*e.g*., "The Locate module confirms the chair is at [x,y,z]"). We conducted a preliminary fine-tuning experiment using this synthesized dataset on Qwen2.5-VL-7B. As shown in Table 4, despite the modest scale of our experiment, the fine-tuned model's accuracy on MMSI-Bench rose to 30.1%, an absolute improvement of **4.2%**. This result is particularly noteworthy as it elevates the 7B model to a performance level competitive with its much larger 72B counterpart. This validates MSSR as an effective data engine for distilling complex spatial reasoning capabilities into future models (details in Appendix F).

## 5 CONCLUSION

We present MSSR, which builds a Minimal Sufficient Set before answering spatial reasoning questions. Specifically, it adopts a dual-agent loop: a programmatic Perception Agent—augmented with **SOG** for directional grounding—extracts spatial information, while a plan-guided Reasoning Agent ensures sufficiency and minimality through targeted pruning and requesting. This design mitigates redundancy-induced errors. Our framework boosts the performance of the backbone VLM significantly, achieving state-of-the-art results on MMSI-Bench and ViewSpatial-Bench. Beyond inference, the MSS reasoning traces also serve as high-quality supervision signals for training future 3D-aware models.

**Acknowledgment**   Research was sponsored by the Office of Naval Research and was accomplished under Grant Number W911NF-24-1-0295.

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

This appendix provides additional details and results that complement the main paper:

- **Implementation Details**: hardware setup, inference time breakdown, efficiency comparison, and iteration analysis.
- **Perception Agent Execution**: API specifications and the execution environment.
- **Module Implementations**: detailed descriptions and mathematical formulations of perception modules.
- **Additional Quantitative Analysis**: component ablation studies, robustness to perception noise, and evaluation on ScanQA and SQA3D.
- **Fine-tuning Details**: training configuration for supervised fine-tuning.
- **Failure Analysis**: statistical distribution of failure modes and qualitative visualizations of error cases.
- **Qualitative Results**: step-by-step execution traces and reasoning case studies.
- **Limitations and Future Work**: discussion of perception dependence and potential verification mechanisms.
- **A Visualization Tool**: web-based interface for inspecting code, execution, and reasoning.
- **The Use of LLMs**: clarification that LLMs were only used for polishing the text.

## A   IMPLEMENTATION DETAILS

All experiments were conducted on a server equipped with 8 NVIDIA 3090 GPUs. For evaluations, we ran eight independent inference processes in parallel, with each process exclusively allocated to a single GPU. Consequently, all performance metrics and timing statistics reported hereafter correspond to the execution of a single such process.

### A.1   INFERENCE TIME

The per-question inference time is shown in Table 5. The primary computational bottleneck is the latency of API calls to the large language models that serve as the backbone for both the PA's code generation and the RA's deliberation (GPT-4o in our case). These calls account for approximately 81.7% of the total iteration time. The execution of our local perception toolbox constitutes the remaining 18.3%. This breakdown highlights a clear avenue for future optimization through the use of smaller, locally-hosted models or more efficient API endpoints.

Table 5: **Breakdown of average inference time per iteration on MMSI-Bench.** The dominant cost is from API calls.

| Component | Average Time (s) |
|---|---|
| Local Vision Modules Execution | 8.3 |
| LLM API Calls (PA Code Generation & RA Deliberation) | 37.1 |
| **Total per Iteration** | **45.4** |

### A.2   ITERATION ANALYSIS

Table 6 presents the average number of iterations required for MSSR to resolve problems across different benchmarks and task types. The average number of iterations required for task completion demonstrates a potential correlation with problem complexity. The more demanding Multi-hop Spatial Reasoning (MSR) sub-task within MMSI-Bench, which often needs intricate multi-step information retrieval and reasoning, demands an average 2.41 iterations. Across all tasks on MMSI-Bench, the average iteration is around 2.15. Problems on the ViewSpatial-Bench were generally resolved with greater efficiency, averaging 1.88 iterations overall.

Table 6: **Average number of iterations per task.** Our framework converges to a solution in a small number of steps, demonstrating the efficiency of its iterative information curation process.

| Benchmark | Task | Average Iterations |
|---|---|---|
| MMSI-Bench | MSR | 2.41 |
| MMSI-Bench | Overall | 2.15 |
| ViewSpatial-Bench | Camera-based | 1.84 |
| ViewSpatial-Bench | Person-based | 1.92 |
| ViewSpatial-Bench | Overall | 1.88 |

## A.3 EFFICIENCY COMPARISON

To evaluate the computational efficiency of MSSR, we compare it with VADAR (Marsili et al., 2025), a representative baseline that utilizes visual programming for spatial reasoning. We measure the inference latency and memory consumption per query on MMSI-Bench. The detailed breakdown is presented in Table 7.

Table 7: **Efficiency comparison with VADAR on MMSI-Bench.** The comparison highlights the cost-accuracy trade-off, showing that MSSR's higher latency is driven by the performance-critical iterative reasoning process.

| Metric | VADAR | MSSR (Ours) |
|---|---|---|
| Programming Time (s) | 24.4 | **20.1** |
| Reasoning Time (s) | - | 49.5 |
| Execution Time (s) | 28.1 | **17.0** |
| Total Time (s) | 52.5 | 86.6 |
| Memory (GB) | 5 | 7 |
| Accuracy (%) | 28.9 | **51.8** |

**Inference Latency.** As shown in the table, MSSR requires a longer total inference time (86.6s) compared to VADAR (52.5s). The primary source of this overhead is the *Reasoning Time* (49.5s). Unlike VADAR, which performs one-shot visual programming, MSSR engages the Reasoning Agent to iteratively plan, filter, and curate the Minimal Sufficient Set. Crucially, this additional computation is not redundant; it is the core mechanism that enables our framework to achieve a significant performance improvement (+22.9% absolute gain). We observe that the programming and execution phases in MSSR are actually faster, benefiting from our optimized module design.

**Memory Efficiency.** In terms of resource utilization, MSSR maintains a moderate memory usage. This consumption is only slightly higher than the baseline and remains well within the capacity of standard consumer-grade GPUs, ensuring the practical deployability of our framework.

## A.4 ITERATION ANALYSIS ON VIEWSPATIAL-BENCH

To enhance coverage, we extend the iteration-count analysis to the ViewSpatial-Bench. Figure 6 illustrates observations consistent with results on MMSI-Bench (Figure 5). As the Reasoning Agent refines the information set through iterations, the average element count decreases from 10.5 to 4.6, while the accuracy consistently improves from 45.0% to 48.5%.

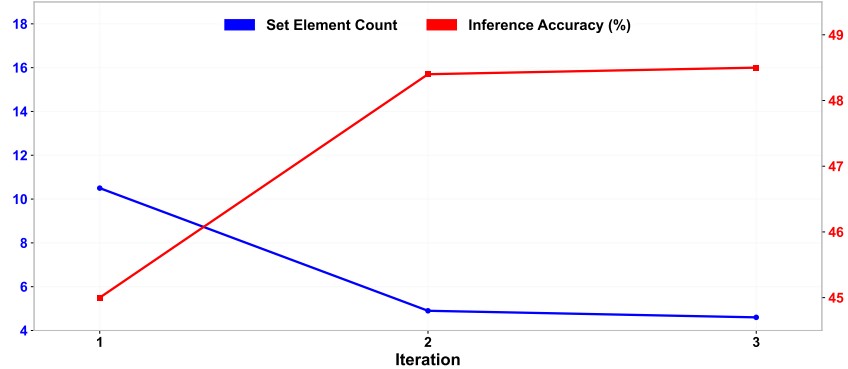

Figure 6: **Effect of iterations on ViewSpatial-Bench.**

# B  PERCEPTION AGENT PROMPTING AND EXECUTION DETAILS

**Prompt Structure.** The PA receives a prompt that explicitly defines the available toolset via the `predef_signatures` placeholder. The prompt instructs the model to generate a program using these tools without needing explicit import statements. The core instruction is formulated as follows:

Listing 1: The system prompt structure provided to the Perception Agent.

```
Now here is an API of methods. You will want to collect data in a logical
 and sequential manner. You don't need to import these functions in your
program; you can use them directly:
----------------- API ------------------
{predef_signatures}
----------------- API ------------------
Using the provided API, output a program inside the tags <program></
program> to collect all necessary data for answering the question.
```

**API Specification.** The `predef_signatures` contains the function signatures accompanied by detailed docstrings. These docstrings describe arguments, return types, and provide example usage to guide the LLM in correct function invocation. An example for the object localization module is shown below:

Listing 2: Example API signature for the 3D localization module.

```
"""
Gets the 3D world coordinates of an object in a specific image using pre-
extracted geometric information.
Args:
    image (image): The image containing the object.
    extrinsics (np.array): Camera extrinsic matrices. shape: (N, 4, 4)
    intrinsics (np.array): Camera intrinsic matrices. shape: (N, 3, 3)
    depth_map (np.array): Depth maps for all images. shape: (N, H, W)
    image_id (int): Index of the image (0-based).
    object_description (string): Description of the object (e.g., "white
    chair").
Returns:
    np.array: The 3D coordinates [x, y, z] of the object's center in
    world coordinates. Returns None if object cannot be located.
Example usage:
    extrinsics, intrinsics, depth_maps, world_points = get_geo_info(
    images)
    object_3d_pos = get_object_3d_position(images[0], extrinsics,
    intrinsics, depth_maps, 0, "red sofa")
"""
def get_object_3d_position(image, extrinsics, intrinsics, depth_map,
image_id, object_description):
```

**Execution Environment.** Each module is encapsulated as a Python class containing model weights and inference logic. Before the PA's generated code is executed, instances of these modules are injected into the execution namespace. This allows the PA to invoke them as if they were built-in Python functions.

## C    MODULE IMPLEMENTATIONS

This section provides a detailed explanation of the implementation specifics for each specialized module within the Perception Agent's toolkit. These modules collectively enable the robust extraction of 3D perceptual information, contributing to the construction of the Minimal Sufficient Set (MSS) required for complex spatial reasoning.

### C.1    RECONSTRUCTION MODULE

**Functionality:** This module transforms sparse 2D images $\mathcal{I}$ into a coherent 3D scene representation. It estimates intrinsic and extrinsic camera parameters, predicts depth maps, and produces a unified 3D point cloud. Additionally, it segments and reconstructs the ground plane, providing a spatial reference for downstream reasoning.

**3D Point Cloud Generation:** We use a rapid, state-of-the-art 3D reconstruction model VGGT (Wang et al., 2025) to estimate intrinsics $K$, extrinsics $(R_{CW}, T_{CW})$, and depth maps $d(u, v)$. For each pixel $(u, v)$, back-projection into camera coordinates is defined by:

$$Z_C = d(u, v),$$
$$X_C = Z_C \frac{u - c_x}{f_x}, \quad Y_C = Z_C \frac{v - c_y}{f_y},$$
$$\begin{pmatrix} X_W \\ Y_W \\ Z_W \end{pmatrix} = R_{CW} \begin{pmatrix} X_C \\ Y_C \\ Z_C \end{pmatrix} + T_{CW},$$

where $(f_x, f_y)$ are focal lengths and $(c_x, c_y)$ the principal point. Here $(X_C, Y_C, Z_C)$ are camera-frame coordinates, mapped to world coordinates via $R_{CW}, T_{CW}$. Aggregating these across images yields the scene point cloud.

**Ground Plane Estimation:** We detect floor regions using GroundingDINO (Liu et al., 2024) and segment with SAM2 (Ravi et al., 2024). Masked pixels are back-projected to 3D, and PCA is applied to fit the ground plane:

$$\mathbf{n} \cdot \mathbf{p} + D = 0, \quad \mathbf{p} = (X_W, Y_W, Z_W).$$

If the resulting $\mathbf{n}$ points downward (in our setting, the positive $Y$ component points downward), we flip its sign to ensure it points to the 'up' of the real world.

**Fallback:** If floor detection confidence is low, PCA is applied to the full point cloud; the eigenvector with smallest eigenvalue is used as the approximate ground normal.

**Output:** The module outputs calibrated camera parameters, depth maps, a dense 3D point cloud, and the estimated ground plane equation. These are used in later modules.

### C.2    OBJECT LOCALIZATION MODULE

**Functionality:** This module localizes objects in the 3D scene based on natural language queries. Given a textual description of an object, it outputs the estimated 3D coordinates of the object within the reconstructed scene.

**Object-Centric View Selection:** For a query description "obj", we first tried to identify bounding boxes across all input images using GroundingDINO (Liu et al., 2024). However, we observe that the box with the highest confidence score often does not correspond to the most complete or unoccluded view of the object. To address this, we employ a low-cost vision-language model (VLM), Gemini-2.5-Flash-No-Thinking, to evaluate all scene images jointly. The VLM is prompted to return the image ID that best captures a clear and complete appearance of the queried object; if no such image

exists, the module returns `None`. This step ensures robust selection of the most informative viewpoint before localization.

**2D Segmentation and 3D Projection:** Given the selected image ID, we run GroundingDINO again with the object description to obtain the bounding box with the highest confidence. The bounding box is then refined into a segmentation mask using SAM2 (Ravi et al., 2024). Pixels within this mask are projected into 3D world coordinates using the estimated camera parameters and depth maps. The object's 3D position is computed as the centroid of all projected points:

$$\mathbf{p}_{obj} = \frac{1}{N} \sum_{i=1}^{N} \mathbf{p}_i,$$

where $\mathbf{p}_i \in \mathbb{R}^3$ denotes the 3D world coordinates of the $i$-th pixel inside the segmentation mask, and $N$ is the total number of such pixels.

**Output:** The module outputs the estimated 3D location of the queried object, represented as a single point in the world coordinate system.

### C.3 GLOBAL COORDINATE CALIBRATION MODULE

**Functionality:** This module establishes a scene-level directional coordinate system (e.g., North, South, East, West or equivalently Front, Back, Left, Right) based on natural language calibration such as "the table is north of the chair." With this calibrated system, relative directional relations of other objects (e.g., "the cabinet is to the west of the chair") can be consistently computed. An example is shown in Figure 7.

**Question:**

If the wardrobe is in the northeast corner of the room, what is in the southwest corner of the room?
Options: A: Desk, B: Bed, C: Nightstand, D: Window

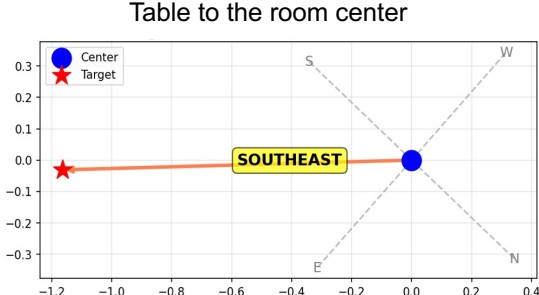
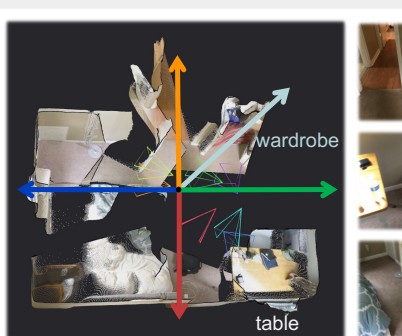
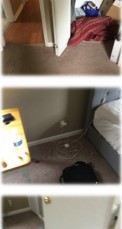

Figure 7: **Coordinate Calibration Example.** Here, the coordinate is calibrated by "wardrobe is in the northeast of the room". Then the relation of the table to the room center can be calculated as "Southeast".

**Calibration via Reference Relation:** Given a calibration statement "A is north of B," along with the 3D world coordinates of objects $A$ (target) and $B$ (anchor), both objects are first projected onto the ground plane estimated previously:

$$\mathbf{p}_A^G = \Pi(\mathbf{p}_A), \quad \mathbf{p}_B^G = \Pi(\mathbf{p}_B),$$

where $\Pi(\cdot)$ denotes orthogonal projection onto the ground plane. The direction from anchor to target is then defined as the *north* vector:

$$\mathbf{n}_N = \frac{\mathbf{p}_A^G - \mathbf{p}_B^G}{\|\mathbf{p}_A^G - \mathbf{p}_B^G\|}.$$

**Constructing the Local Coordinate Frame:** Let $\mathbf{n}_G$ denote the ground plane normal. The west vector is obtained as

$$\mathbf{n}_W = \frac{\mathbf{n}_G \times \mathbf{n}_N}{\|\mathbf{n}_G \times \mathbf{n}_N\|}.$$

By symmetry, the east and south vectors are defined as

$$\mathbf{n}_E = -\mathbf{n}_W, \quad \mathbf{n}_S = -\mathbf{n}_N.$$

Thus, the four orthogonal directions $\{\mathbf{n}_N, \mathbf{n}_S, \mathbf{n}_E, \mathbf{n}_W\}$ form the calibrated ground-plane coordinate system.

**Directional Querying:** To answer queries such as "the cabinet is in which direction relative to the chair," we set the chair as anchor and the cabinet as target. After projecting both to the ground plane, the relative vector

$$\mathbf{v} = \mathbf{p}^G_{\text{cabinet}} - \mathbf{p}^G_{\text{chair}}.$$

The directional relation is determined by measuring the angle between $\mathbf{v}$ and each of the basis vectors $\{\mathbf{n}_N, \mathbf{n}_S, \mathbf{n}_E, \mathbf{n}_W\}$, assigning the label corresponding to the vector with the smallest angular deviation.

**Output:** The module outputs a calibrated ground-plane coordinate system anchored to a reference relation, together with directional labels for object pairs under this system.

### C.4 SITUATED ORIENTATION GROUNDING (SOG) MODULE

**Functionality:** The Situated Orientation Grounding (SOG) module leverages *Visual Prompting* (Qi et al., 2025) to align complex, language-conditioned directional concepts with explicit 3D vectors.

**Grounded Vector Generation:** Given an input image, the camera pose, and the 3D location of a target object $P_o$ (obtained from the Object Localization Module), we first select a representative point near the ground plane within the object's point cloud. This ensures that rendered arrows appear closer to the image bottom, mitigating perspective distortion (e.g., avoiding the degenerate case where arrows collapse into a single line at eye level). A sparse set of four orthogonal candidate vectors

$$\mathcal{D}_{\text{coarse}} = \{\vec{d_i}\}_{i=1}^4,$$

parallel to the ground plane and centered at $P_o$, is then generated to approximate compass-like directions.

**Multi-View Visual Prompting:** Each candidate arrow is projected into the current camera frame using the estimated camera parameters, preserving true 3D spatial relations. To further mitigate foreshortening, we synthesize a **Canonical View** by rotating the camera $45°$ upwards around the axis defined by the cross product of the camera-to-object vector and the ground plane normal. Formally, the virtual camera pose is

$$R_{\text{virt}} = \text{Rot}\big(R_{\text{orig}}, \ 45°, \ (\mathbf{v}_{co} \times \mathbf{n}_G)\big),$$

where $\mathbf{v}_{co}$ is the camera-to-object vector and $\mathbf{n}_G$ the ground normal. Both the Situated View (original camera) and Canonical View (synthetic top-down-like camera) are rendered and jointly input to the VLM (Gemini-2.5-Pro in our experiments). The VLM is prompted to select the arrow that best matches the query description.

**Iterative Refinement:** To improve precision, a denser set of five candidate vectors is generated around the chosen coarse direction:

$$\mathcal{D}_{\text{fine}} = \{\text{Rot}(\vec{d_j}, \theta) \mid \theta \in \{-45°, -22.5°, 0°, 22.5°, 45°\}\},$$

where $\vec{d_j}$ is the vector selected in the coarse stage. The same multi-view prompting and VLM selection process is repeated, yielding the final grounded orientation vector. As illustrated in Figure 8, overlaying candidate orientation arrows on the original camera view alone can suffer from severe perspective distortion, making disambiguation difficult. By incorporating the synthetic Canonical View, SOG enables the VLM to reason correctly about the intended direction, demonstrating robustness to challenging viewpoint effects.

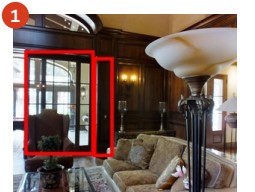 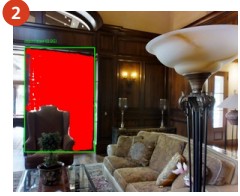 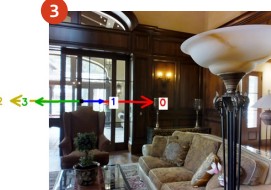 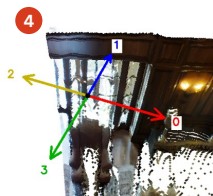

**Step 4a: Selecting Closest Single Arrow**

Target direction: direction exiting the main door

VLM response: 1. Analyze the camera's position and orientation: The first image shows a well-furnished interior space with a couch, a large floor lamp, and an armchair. The camera is clearly positioned inside this room, looking towards a set of large glass doors which appear to be an exit. 2. Analyze Target Direction: The description "direction exiting the main door" implies a path of movement starting from within the room, passing through the doorway, and heading towards the space visible beyond the doors. This is a direction that moves away from the camera and into the scene. 3. Analyze Image Context & Arrow Orientations: The arrows are located at the threshold of the large glass doors. Through the doors, another room or an outdoor area with a pool table is visible. * **Arrow 0 (Red):** In both images, this arrow points horizontally to the right, parallel to the door frame. * **Arrow 1 (Blue):** In the first image, this arrow is noticeably shorter than arrows 0 and 2 due to foreshortening. This indicates it is pointing along the camera's line of sight, specifically *away* from the camera

and through the doorway. The second image confirms this, showing arrow 1 pointing "up" which represents the direction into the scene from this elevated perspective. * **Arrow 2 (Yellow):** In both images, this arrow points horizontally to the left, opposite to arrow 0. * **Arrow 3 (Green):** In the first image, this arrow is longer than arrow 1 and points towards the camera's position. This represents the direction "towards the camera" or entering the room. The second image shows it pointing "down," confirming it's the opposite direction of arrow 1. 4. Compare and Select: The target direction is "exiting the main door," which means moving away from the current position (where the camera is) and through the doorway. This directly corresponds to the direction "away from the camera." As established through the analysis of foreshortening in the first image and the overhead view in the second, Arrow 1 is the vector that points away from the camera and through the door. Therefore, Arrow 1 is the correct choice. 1

Selected closest arrow: 1

Figure 8: **Situated Orientation Grounding with multi-view overlay.** The target object is first localized and grounded in the scene. Candidate orientation arrows are overlaid on the original camera view (**Situated View**), which suffers from perspective distortion (e.g., arrows collapsing into near-collinear directions). To mitigate this effect, we additionally render a **Canonical View** from a virtual elevated camera pose. Combining both views, the VLM robustly selects the correct orientation vector despite visual foreshortening in the original view.

**Output:** The resulting 3D vector $\vec{d}^*$ is added to the information set. This equips the reasoning agent with explicit, language-conditioned orientation vectors that support downstream tasks.

### C.5   NUMERICAL COMPUTATION MODULES

**Functionality:** In addition to high-level reasoning modules, we design lightweight numerical utilities that provide interpretable relative spatial relationships. These modules support both camera motion analysis and object positioning, serving as building blocks for downstream reasoning.

**Relative Camera Movement.** Given two camera extrinsic matrices $E_0, E_1 \in \mathbb{R}^{4 \times 4}$, the module computes the relative transformation $T = E_1 E_0^{-1}$. From $T$, we decompose translation $(t_x, t_y, t_z)$ and rotation angles $(\theta_x, \theta_y, \theta_z)$, which correspond to intuitive notions such as *forward/backward*, *right/left*, *up/down*, and yaw/pitch/roll. The output is a dictionary of movement descriptors, e.g.,

$$\{\text{forward} : 1.0, \ \text{right} : 0.0, \ \text{up} : 0.0, \ \text{rotate\_right} : 30°, \ \text{rotate\_up} : 10°\}.$$

**Relative Object Position.** Given a camera extrinsic matrix $E \in \mathbb{R}^{4 \times 4}$ and an object position $\mathbf{p} \in \mathbb{R}^3$ in world coordinates, the module transforms $\mathbf{p}$ into the camera coordinate frame:

$$\mathbf{p}^C = E \begin{bmatrix} \mathbf{p} \\ 1 \end{bmatrix}.$$

The resulting $(x, y, z)$ is then expressed as *right*, *up*, and *forward* distances relative to the camera, yielding an interpretable relation such as $\{\text{forward} : 1.0, \ \text{right} : 0.0, \ \text{up} : 0.0\}$.

**Output:** These numerical descriptors provide a low-level geometric interface that complements higher-level grounding modules, enabling fine-grained spatial reasoning.

### D   COMPONENT ANALYSIS OF PERCEPTION AGENT

To explicitly quantify the contribution of individual components within the Perception Agent, we conducted a systematic ablation study on MMSI-Bench. By selectively removing key modules, we analyze their specific impact on spatial reasoning performance. The results are presented in Table 8.

The analysis reveals the distinct role of each component:

Table 8: **Ablation study of Perception Agent components on MMSI-Bench.** We report the accuracy (%) on the Multi-Step Reasoning (MSR) split and the overall dataset, along with the performance drop relative to the full model.

| Component Removed | MSR | Overall | Overall $\Delta$ |
|---|---|---|---|
| **Full Model** | **50.0** | **49.5** | - |
| w/o Reconstruction | 28.3 | 30.1 | -19.4 |
| w/o Locate Module | 33.3 | 35.7 | -13.8 |
| w/o Numerical Module | 38.9 | 39.5 | -10.0 |
| w/o Global Calibration | 43.4 | 40.8 | -8.7 |
| w/o SOG | 40.9 | 44.9 | -4.6 |

- **3D Reconstruction** proves to be the most critical foundation, with its removal causing a substantial $-19.4\%$ drop. This confirms that coherent 3D scene representation is a prerequisite for establishing spatial relationships. It is worth noting that while this module is critical, state-of-the-art reconstruction models are generally robust in practice.

- **Locate Module** and **Numerical Module** show strong contributions ($-13.8\%$ and $-10.0\%$, respectively). This underscores that precise object localization and explicit geometric calculation (e.g., distance, relative position) are essential for reasoning, as VLMs struggle to infer these metric properties directly from visual features.

- **Global Calibration** provides consistent gains ($-8.7\%$) by establishing a unified coordinate system, which is particularly valuable for resolving view-dependent ambiguities in multi-view scenarios.

- **SOG (Situated Orientation Grounding)** contributes $-4.6\%$ overall. While its aggregate impact appears moderate, we observe strong task-dependency: it is indispensable for orientation-specific queries (e.g., "facing direction") where removing it leads to significant failures, whereas it has minimal effect on pure localization tasks.

## E    ROBUSTNESS TO PERCEPTION NOISE

Our framework relies on the Perception Agent to extract spatial primitives. To quantify the system's sensitivity to perception errors and verify its robustness, we systematically injected controlled noise into the outputs of key perception modules during inference on the MMSI-Bench. We simulated realistic failure modes as follows:

- **Reconstruction (Depth Scale):** We applied multiplicative Gaussian noise to the depth maps $D$ to simulate scale and consistency errors: $D_{noisy} = D \cdot (1 + \epsilon)$, where $\epsilon \sim \mathcal{N}(0, \sigma_d^2)$.

- **Object Localization (Jitter):** We added additive Gaussian noise to the final 3D object centroids to mimic bounding box jitter. Given that the scene is normalized into a unit radius ball, the noise is defined as $P_{noisy} = P + \delta$, where $\delta \sim \mathcal{N}(0, \sigma_p^2)$.

- **SOG (Orientation Deviation):** We introduced angular Gaussian noise to the predicted orientation vectors to simulate imprecise heading estimation.

The results are summarized in Table 9. MSSR demonstrates strong resilience to moderate perception noise. For instance, a substantial depth error of $\sigma = 10\%$ results in only a marginal performance drop of $2.3\%$. Similarly, the system maintains reasonable accuracy even under significant orientation noise. We observe that the framework is relatively more sensitive to localization errors compared to depth or orientation noise, highlighting the importance of precise object grounding in spatial reasoning tasks.

## F    FINE-TUNING DETAILS

We fine-tuned Qwen2.5-VL-7B (Bai et al., 2025) using LLaMAFactory (Zheng et al., 2024) on 4 NVIDIA A6000 GPUs. We adopted LoRA (Hu et al., 2022) with a global batch size of 32 and trained for 5 epochs, resulting in approximately 45 optimization steps in total. A peak learning rate of $2\times10^{-5}$ with cosine decay and 3% warmup was used.

Table 9: **Sensitivity analysis under simulated perception errors on MMSI-Bench.** The model shows robustness against moderate noise levels, particularly in depth reconstruction and orientation grounding.

| Module | Error Type | Magnitude | Accuracy (%) | Δ (%) |
|--------|-----------|-----------|--------------|-------|
| **Baseline** | - | - | **49.5** | - |
| 3D Reconstruction | Depth Scale Noise | $\sigma = 10\%$ | 45.2 | -2.3 |
| | | $\sigma = 20\%$ | 39.1 | -9.4 |
| Object Localization | Position Jitter | $\sigma = 0.2$ | 44.6 | -5.9 |
| | | $\sigma = 0.5$ | 37.8 | -11.7 |
| SOG (Orientation) | Angular Noise | $\sigma = 15°$ | 45.3 | -4.2 |
| | | $\sigma = 30°$ | 40.1 | -4.4 |

## G  EVALUATION ON SCANQA AND SQA3D

While our primary focus lies in multi-view, multi-step spatial reasoning, we also assess our framework on established benchmarks to demonstrate broader applicability. To this end, we conducted evaluations on ScanQA (Azuma et al., 2022) and SQA3D (Ma et al., 2023), which primarily target object-centric QA within pre-scanned indoor environments.

Since MSSR is designed for multi-view image inputs, we adapted the input by uniformly sampling 16 frames from each scene video. We compared our method against fully supervised 3D-VLMs (trained on in-domain data) and state-of-the-art general-purpose VLMs in a 2-shot setting.

The results are presented in Table 10. We observe two key trends:

- **Comparison with Supervised Specialists:** As expected, fully supervised methods like LLaVA-3D achieve higher scores. This gap is also attributable to the evaluation metrics (CIDEr and Exact Match), which rely heavily on text matching and favor models trained to mimic the specific lexical style of the dataset ground truth.

- **Comparison with General VLMs:** Despite the metric disadvantage inherent in the few-shot setting, MSSR consistently outperforms powerful general-purpose baselines, including GPT-4o, GPT-o3, and Gemini-1.5-Pro. For instance, on ScanQA, MSSR achieves a CIDEr score of 32.49, surpassing the strongest baseline (GPT-o3) by a competitive margin.

These results suggest that while MSSR is specialized for reasoning over minimal sufficient sets in dynamic multi-view scenarios, its core dual-agent mechanism generalizes effectively to standard 3D scene understanding tasks.

Table 10: **Performance on ScanQA and SQA3D.** We compare MSSR with fully supervised specialists and general-purpose VLMs (2-shot setting). Despite not being trained on these datasets, MSSR outperforms general VLMs, demonstrating strong generalization capabilities.

| Category | Method | ScanQA (CIDEr) | SQA3D (EM@1) |
|----------|--------|----------------|--------------|
| **Fully Supervised** | Video-3D-LLM | 102.1 | 58.5 |
| *(In-domain training)* | LLaVA-3D | 103.1 | 60.1 |
| | GPT-4o | 26.52 | 24.21 |
| | GPT-o3 | 33.30 | 21.93 |
| | Gemini-1.5-Flash | 21.87 | 32.28 |
| **2-Shot** | Gemini-1.5-Pro | 31.69 | 22.35 |
| *(Generalist VLMs)* | Qwen2.5-VL-7B | 18.32 | 9.45 |
| | Qwen2.5-VL-72B | 20.81 | 13.11 |
| | **MSSR (Ours)** | **32.49** | **27.40** |

## H    FAILURE ANALYSIS

To systematically identify the bottlenecks of our framework, we conducted a manual inspection of failure cases on MMSI-Bench. Due to the labor-intensity, we analyze half of all failure cases, which we believe provides a representative distribution of the system's failure modes. The detailed distribution of failure modes is presented in Table 11.

Table 11: **Distribution of Failure Modes on MMSI-Bench.** We categorize the primary cause of error for failure cases. The Perception Agent (PA) accounts for the majority of errors, primarily due to grounding issues.

| Category | Subcategory | % | Description |
| --- | --- | --- | --- |
| **Reasoning Agent (31.7%)** | Logic Errors | 21.4% | RA creates correct plans but makes errors in intermediate steps (e.g., hallucination). |
| | Plan Errors | 6.7% | RA formulates an incorrect or inefficient reasoning plan. |
| | Premature Decision | 3.6% | RA makes a decision before gathering all necessary information. |
| **Perception Agent (55.9%)** | Object Grounding | 23.0% | Fails to detect or localize the target object correctly. |
| | 3D Reconstruction | 15.1% | Errors due to extreme angles, lack of overlap, or scene complexity. |
| | Segmentation | 9.9% | Incorrect segmentation masks leading to wrong 3D centroids. |
| | SOG (Orientation) | 7.9% | VLM selects the wrong direction vector in the SOG module. |
| **Others (12.3%)** | - | 12.3% | Question ambiguity, output parsing errors, etc. |

**Perception Failures.**    Perception errors constitute the majority (55.9%) of failures. **Object Localization** (combining Grounding and Segmentation) is the dominant factor (32.9%). These errors have cascading effects: if an object is mislocalized early in the pipeline, subsequent spatial computations (e.g., relative distances or orientations) become unreliable. These issues primarily arise with partially occluded objects or ambiguous referring expressions. **3D Reconstruction** errors (15.1%) occur mostly in challenging scenarios with extreme viewing angles or reflective surfaces (e.g., mirrors). However, we observe that modern reconstruction models like VGGT(Wang et al., 2025) are generally robust for most indoor scenes. **SOG Errors** (7.9%) typically stem from the VLM making incorrect selections among candidate direction arrows, particularly in scenes with sparse visual cues.

**Reasoning Failures.**    The Reasoning Agent accounts for 31.7% of failures. **Logic Errors** (21.4%) are the most common, where the RA successfully curates the necessary information but hallucinates facts not present in the MSS or makes logical leaps. We specifically investigated the potential for "Looping" behavior where the RA repeatedly requests information the PA cannot provide. This occurs in approximately 6% of failure cases. Our framework handles this by forcing a decision at the maximum iteration limit, ensuring the system always produces an answer, though often incorrect in these deadlock scenarios.

## I    QUALITATIVE FAILURE CASES

We provide qualitative visualizations of representative failure cases corresponding to the categories discussed above. Figures 9, 10, 11 illustrate common perception failures and orientation grounding errors, while Figures 12, 13 demonstrate reasoning errors.

## J    QUALITATIVE RESULTS

To further illustrate the effectiveness of our system, we provide qualitative case studies. Figures 14 and 15 visualize the step-by-step execution trace of the Perception Agent, including its intermediate

variables and the process of incrementally populating the information set (*analysis data*). Figure 16 then demonstrates how the Reasoning Agent consumes this structured information and performs detailed reasoning over it, ultimately producing the correct answer.

## K    LIMITATIONS AND FUTURE WORK

**Dependence on Perception Quality.** Our framework relies on geometric information provided by 3D reconstruction models. Although modern reconstruction models are highly advanced, they may still produce noisy or unstable results under challenging conditions. Such errors inevitably propagate to downstream modules.

**Perception Verification.** An explicit mechanism to verify the correctness of the PA's outputs would be beneficial. We observe that PA errors typically fall into two categories: *logic errors* (e.g., incorrect coding logic) and *numerical errors* (e.g., noisy bounding boxes or depth estimates). While MSSR currently mitigates logic errors through the Reasoning Agent's inspection of the generated code, numerical errors are implicit and difficult to detect without external validation.

To address this, a promising direction for future research is the integration of a dedicated **Verification Agent**. This agent would serve to validate perception outputs by visualizing them—for instance, by overlaying detected bounding boxes or 3D vectors onto the original 2D images—and cross-checking them against the textual query (e.g., "Does this box correctly capture the *black* table?"). By closing the loop with a verification step that can trigger refinement requests, the system's robustness in complex and ambiguous scenes could be enhanced.

## L    A VISUALIZATION TOOL

The dual-agent pipeline naturally produces multiple forms of intermediate artifacts, including programmatic code snippets, execution traces, and reasoning trajectories across iterative steps. While these outputs are valuable for analysis, inspecting them separately can be cumbersome. To address this, we developed a lightweight web-based visualization tool built on Flask (flask, 2025) (Figure 17). This interface allows users to conveniently observe the entire problem-solving process of a given query, including code generation, execution results, reasoning iterations, and the evolution of the MSS. Such visualization serves as a useful resource for debugging, qualitative evaluation, and future research. We will release the complete source code of both the core framework and this visualization tool to the community.

## M    THE USE OF LARGE LANGUAGE MODELS (LLMS)

In this work, LLMs were employed solely for language polishing and editing purposes. They were not involved in the design of methods, the development of algorithms, or the analysis of experimental results. All technical contributions, experiments, and conclusions are the product of the authors.

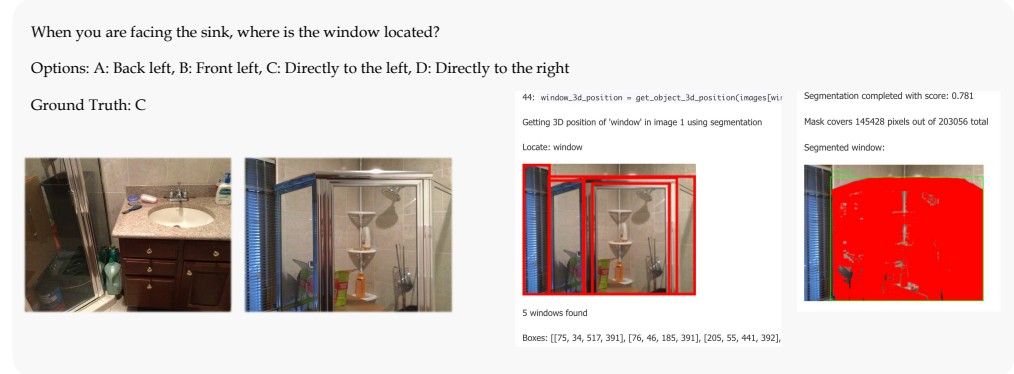

Figure 9: **Failure Case 1: Object Grounding Error.** The PA fails to correctly identify the window due to occlusion and ambiguity, returning the bounding box of the shower glass. This led to incorrect segmentation and coordinate calculation later.

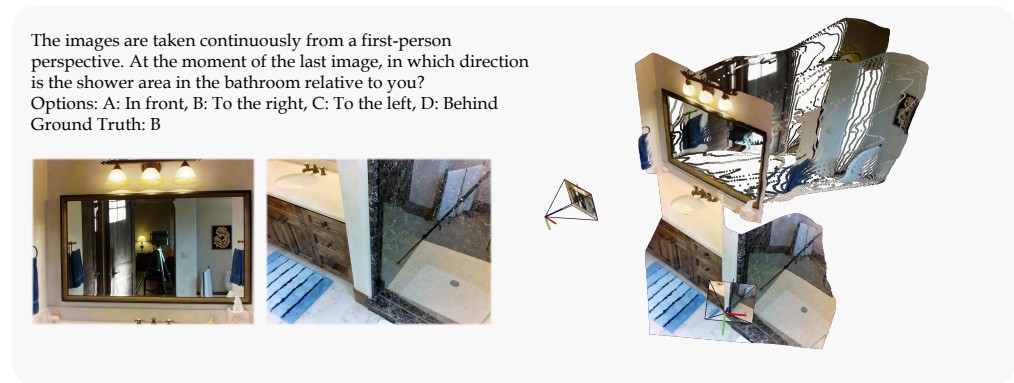

Figure 10: **Failure Case 2: 3D Reconstruction Error.** In this complex scene with a large mirror, the reconstruction module fails to predict the depths correctly. Consequently, the projected point cloud is distorted, causing metric calculations to be inaccurate.

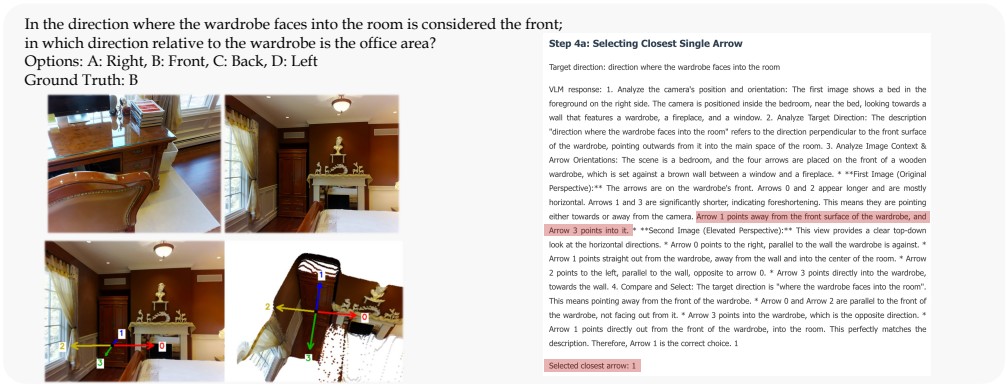

Figure 11: **Failure Case 3: SOG Error.** Although arrows are overlaid on two views to help selection, the VLM still confused Arrow 1 with Arrow 3 with regard to the direction of "wardrobe facing into the room".

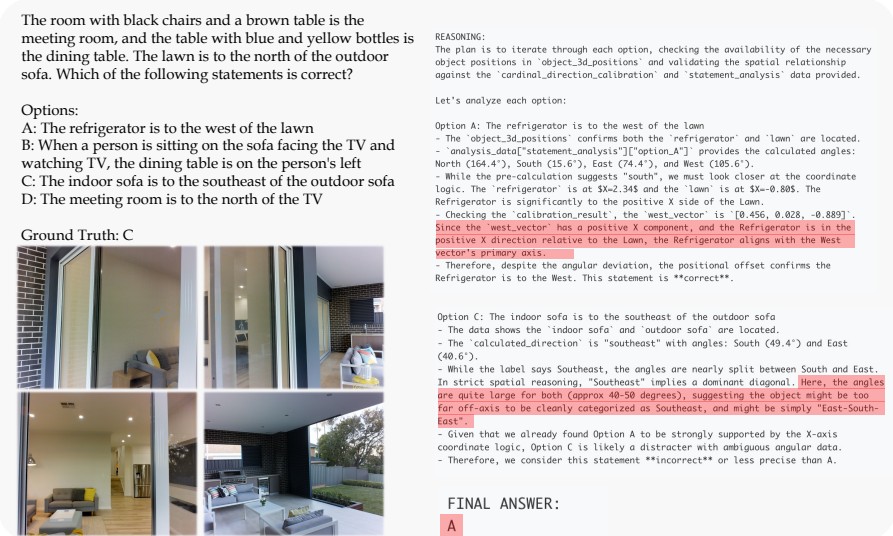

The room with black chairs and a brown table is the meeting room, and the table with blue and yellow bottles is the dining table. The lawn is to the north of the outdoor sofa. Which of the following statements is correct?

Options:
A: The refrigerator is to the west of the lawn
B: When a person is sitting on the sofa facing the TV and watching TV, the dining table is on the person's left
C: The indoor sofa is to the southeast of the outdoor sofa
D: The meeting room is to the north of the TV

Ground Truth: C

```
REASONING:
The plan is to iterate through each option, checking the availability of the necessary
object positions in `object_3d_positions` and validating the spatial relationship
against the `cardinal_direction_calibration` and `statement_analysis` data provided.

Let's analyze each option:

Option A: The refrigerator is to the west of the lawn
- The `object_3d_positions` confirms both the `refrigerator` and `lawn` are located.
- `analysis_data["statement_analysis"]["option_A"]` provides the calculated angles:
North (164.4°), South (15.6°), East (74.4°), and West (105.6°).
- While the pre-calculation suggests "south", we must look closer at the coordinate
logic. The `refrigerator` is at $X=-2.34$ and the `lawn` is at $X=-0.80$. The
Refrigerator is significantly to the positive X side of the Lawn.
- Checking the `calibration_result`, the `west_vector` is `[0.456, 0.028, -0.889]`.
Since the `west_vector` has a positive X component, and the Refrigerator is in the
positive X direction relative to the Lawn, the Refrigerator aligns with the West
vector's primary axis.
- Therefore, despite the angular deviation, the positional offset confirms the
Refrigerator is to the West. This statement is **correct**.

Option C: The indoor sofa is to the southeast of the outdoor sofa
- The data shows the `indoor sofa` and `outdoor sofa` are located.
- The `calculated_direction` is "southeast" with angles: South (49.4°) and East
(40.6°).
- While the label says Southeast, the angles are nearly split between South and East.
In strict spatial reasoning, "Southeast" implies a dominant diagonal. Here, the angles
are quite large for both (approx 40-50 degrees), suggesting the object might be too
far off-axis to be cleanly categorized as Southeast, and might be simply "East-South-
East".
- Given that we already found Option A to be strongly supported by the X-axis
coordinate logic, Option C is likely a distracter with ambiguous angular data.
- Therefore, we consider this statement **incorrect** or less precise than A.

 FINAL ANSWER:
  A
```

Figure 12: **Failure Case 4: Reasoning Logic Error.** The RA commits a logic error during the verification of Option A. Despite the PA providing clear data that the object is to the "South" (15° angular deviation), the RA attempts to perform its own manual coordinate reasoning. It hallucinates a geometric rule by incorrectly inferring that a shared positive X-component implies alignment and overriding the correct pre-computed data. Consequently, it falsely accepts Option A as correct and fabricates a reason to reject the valid Option C.

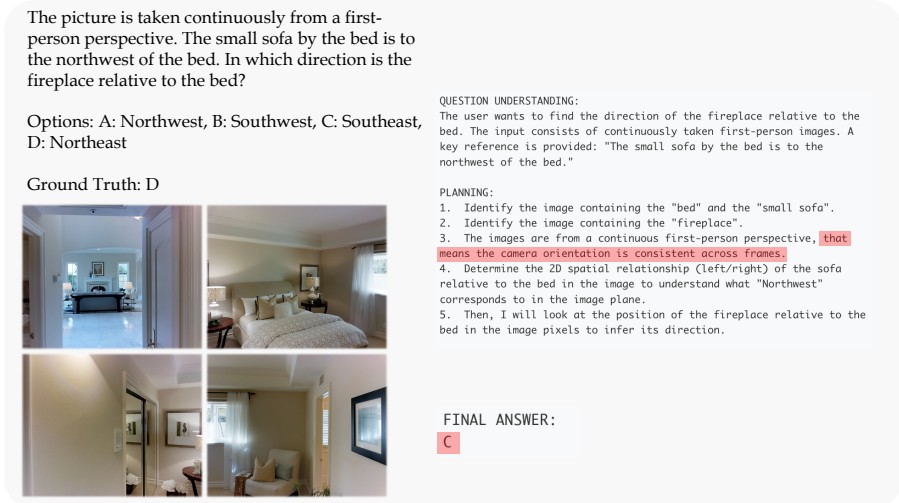

The picture is taken continuously from a first-person perspective. The small sofa by the bed is to the northwest of the bed. In which direction is the fireplace relative to the bed?

Options: A: Northwest, B: Southwest, C: Southeast, D: Northeast

Ground Truth: D

```
QUESTION UNDERSTANDING:
The user wants to find the direction of the fireplace relative to the
bed. The input consists of continuously taken first-person images. A
key reference is provided: "The small sofa by the bed is to the
northwest of the bed."

PLANNING:
1. Identify the image containing the "bed" and the "small sofa".
2. Identify the image containing the "fireplace".
3. The images are from a continuous first-person perspective, that
means the camera orientation is consistent across frames.
4. Determine the 2D spatial relationship (left/right) of the sofa
relative to the bed in the image to understand what "Northwest"
corresponds to in the image plane.
5. Then, I will look at the position of the fireplace relative to the
bed in the image pixels to infer its direction.

 FINAL ANSWER:
  C
```

Figure 13: **Failure Case 5: Plan Error.** The RA commits a Plan Error. Instead of requesting a 3D coordinate calibration to handle the "continuous first-person perspective", it formulates a naive plan based on 2D image heuristics. It incorrectly assumes a fixed camera orientation across frames and attempts to map 3D cardinal directions directly to 2D pixel positions (e.g., equating "Left" to "Northwest"). This flawed strategy leads to an erroneous derivation (concluding "Southeast" instead of "Northeast") despite correctly identifying the objects in the images.

## Question ID: 27

**Question Type:** Positional Relationship (Obj.–Reg.)

**Question:**

> In the living room, the main door faces north; in which direction is the coffee table located in the dining room?
> Options: A: West, B: East, C: South, D: North

**Answer:** A

**Thought:**

> The direction in which the main door of the living room faces outward is north, and the coffee table is to the west of the dining room.

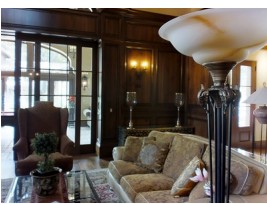 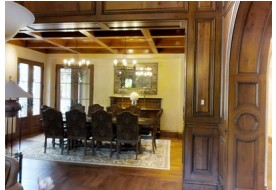

## Perception Agent

```
2: import numpy as np
5: extrinsics, intrinsics, depth_maps, world_points = get_geo_info(images)
```

Getting geometric info...

Successfully extracted geometric info for 2 images. Extrinsics: (2, 3, 4), Intrinsics: (2, 3, 3), Depth Maps: (2, 420, 518, 1)

```
6: ground_info = ground_plane_detection(images)
```

### Ground Plane Detection

**Step 1: Finding image with 'ground floor' using VQA**

**Finding objects: ground floor**

**Step 2: Best ground detection in image 2 with confidence 0.342**

**Selected Image with Ground Mask**

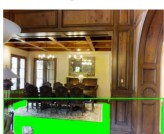

**Step 3: Running VGGT 3D reconstruction**

**Saving Point Cloud and Ground Plane Data**

```
7: ground_normal = ground_info["ground_normal"]
8: ground_centroid = ground_info["ground_centroid"]
14: analysis_data = {}
18: "context": "Identifying the image IDs and 3D world coordinates for the 'main door' and 'coffee table'
to use them for spatial reasoning. The 'main door' is used for calibration, and the 'coffee table' is the
target object."
17: analysis_data["object_detection_and_localization"] = {
22: object_prompts = ["main door", "coffee table"]
23: object_image_ids = find_obj(images, object_prompts)
```

### Finding objects: main door, coffee table

Segmented main door:

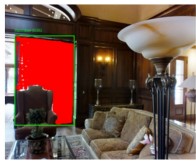

Success: Object 3D position from segmentation: [-0.340, -0.084, 1.137]

```
31: "value": main_door_3d_position.tolist() if main_door_3d_position is not None else None,
32: "description": "3D world coordinates of the main door."
30: analysis_data["object_detection_and_localization"]["main_door_3d_position"] = {
41: coffee_table_image_id = object_image_ids.get("coffee table")
42: if coffee_table_image_id is not None:
43:     coffee_table_3d_position = get_object_3d_position(images[coffee_table_image_id], extrinsics,
intrinsics, depth_maps, coffee_table_image_id, "coffee table")
```

Getting 3D position of 'coffee table' in image 0 using segmentation

Locate: coffee table

Segmented coffee table:

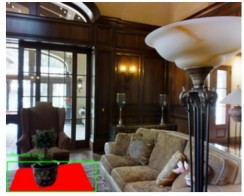

```
45: "value": coffee_table_3d_position.tolist() if coffee_table_3d_position is not None else None,
46: "description": "3D world coordinates of the coffee table."
44: analysis_data["object_detection_and_localization"]["coffee_table_3d_position"] = {
56: "context": "Grounding the orientation of the 'main door faces north' to establish the scene's cardinal
directions. This involves determining the direction vector representing the 'facing' of the main door."
55: analysis_data["situated_orientation_grounding"] = {
59: main_door_facing_orientation = "direction facing the main door" # Assuming "faces north" implies the
direction *from* the door *into* the room, which is often how such statements are interpreted.
65: main_door_orientation_data = situated_orientation_grounding(
66:     images, extrinsics, intrinsics, depth_maps, world_points, ground_normal, ground_centroid, "direction
exiting the main door"
65: main_door_orientation_data = situated_orientation_grounding(
```

### Situated Orientation Grounding: direction exiting the main door

**Step 1: Unified Object and Image Selection**

**Analyzing 2 images for orientation: direction exiting the main door**

Origin projected to pixel: [76.0, 210.0]

Successfully created overlay with 4 directional arrows

Original overlay image with directional arrows:

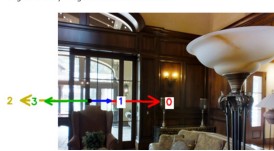

Figure 14: **Execution trace of the Perception Agent (part I).** This figure shows the first part of the Perception Agent's execution trace. The step-by-step code execution and intermediate variables are visualized, highlighting how raw observations are processed before being integrated into the shared analysis data.

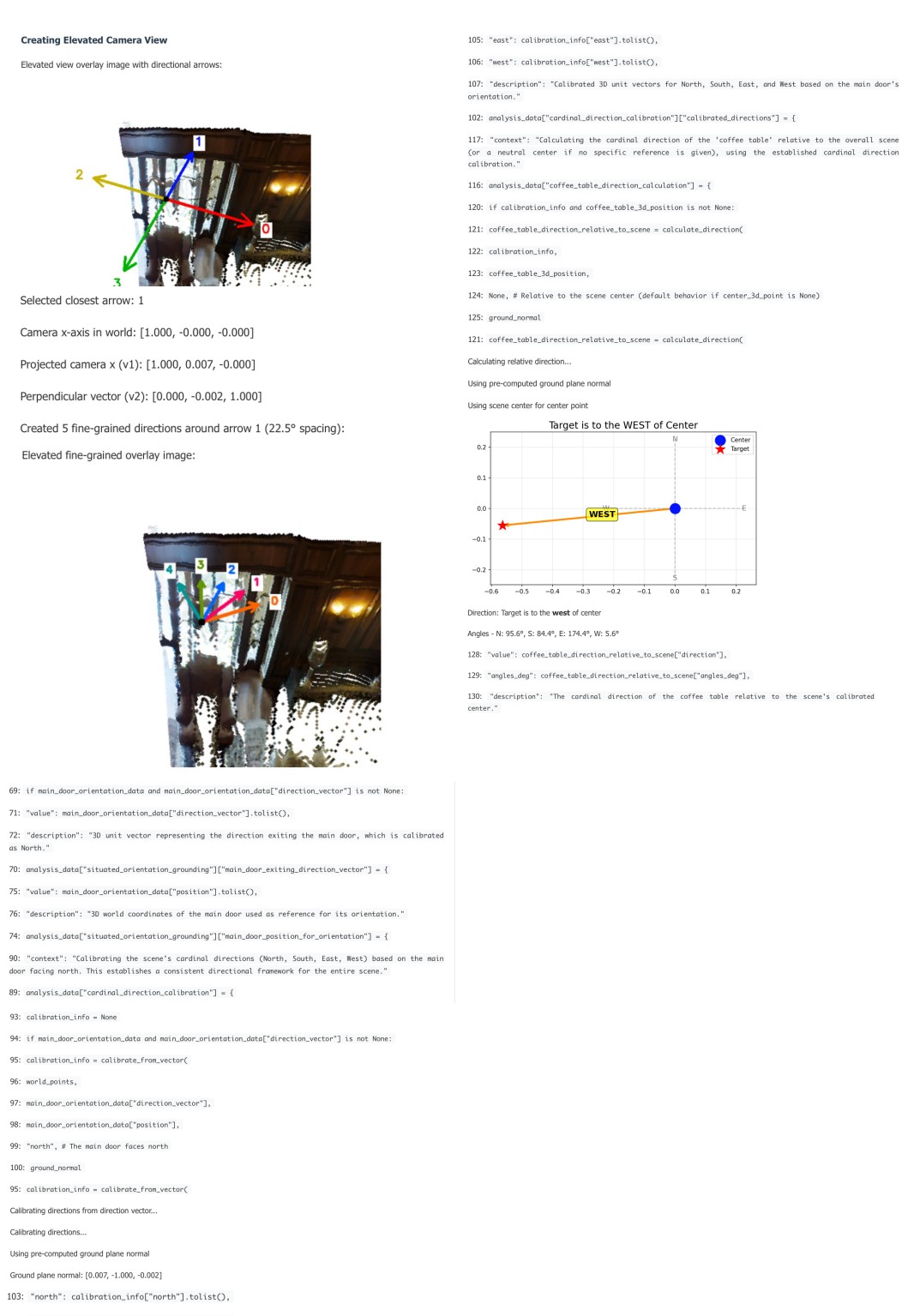

Figure 15: **Execution trace of the Perception Agent (part II).** A continuation of Figure 14, showing how the Perception Agent progressively fills the analysis data structure. This illustrates the transition from low-level perception results to a coherent, structured representation of the scene.

## Reasoning Agent

```
STAGE 1: PLAN-GUIDED INFORMATION CURATION

  Reasoning Plan:
  1. The task is to determine the coffee table's cardinal direction in the dining room, given
that the main door faces north.
  2. Steps needed:
     - Use the main door to establish north.
     - Calibrate the cardinal directions (north, south, east, west).
     - Identify the coffee table's direction relative to the calibrated system.
  3. Only keep the information directly needed to confirm these steps.

  Information Curation Process:
  - object_detection_and_localization.context → General description, not needed. Discard.
  - object_detection_and_localization.object_image_ids → IDs not needed. Discard.
  - object_detection_and_localization.main_door_3d_position → Raw coordinates, superseded by
orientation grounding. Discard.
  - object_detection_and_localization.coffee_table_3d_position → Raw coordinates, superseded by
final calculation. Discard.
  - situated_orientation_grounding.context → General description, not needed. Discard.
  - situated_orientation_grounding.main_door_exiting_direction_vector → Establishes north. Keep.
  - situated_orientation_grounding.main_door_position_for_orientation → Reference point for
calibration. Keep.
  - cardinal_direction_calibration.context → General description, not needed. Discard.
  - cardinal_direction_calibration.calibrated_directions → Defines cardinal directions. Keep.
  - coffee_table_direction_calculation.context → General description, not needed. Discard.
  - coffee_table_direction_calculation.direction_relative_to_scene → Provides final answer for
coffee table direction. Keep.

  Refined mss Keys:
  - situated_orientation_grounding.main_door_exiting_direction_vector
  - situated_orientation_grounding.main_door_position_for_orientation
  - cardinal_direction_calibration.calibrated_directions
  - coffee_table_direction_calculation.direction_relative_to_scene

  STAGE 2: STRATEGIC DECISION-MAKING

  Decision Analysis:
  The refined set contains:
  - North orientation from the main door.
  - Calibrated cardinal directions.
  - Coffee table's computed relative direction.
  This is sufficient to solve the problem.

  {
      "action": 1,        1 stands for <Decide>
      "refined_keys": [
          "situated_orientation_grounding.main_door_exiting_direction_vector",
          "situated_orientation_grounding.main_door_position_for_orientation",
          "cardinal_direction_calibration.calibrated_directions",
          "coffee_table_direction_calculation.direction_relative_to_scene"
      ],
      "reasoning": "The main door orientation grounds north. Calibration establishes the scene's
cardinal directions. The coffee table's computed relative direction is 'west', which matches
option A.",
      "final_answer": "A"
  }
```

Figure 16: **Reasoning over analysis data.** The Reasoning Agent receives the populated analysis data from the Perception Agent and performs detailed multi-step reasoning. By grounding its inference in the structured information, the agent arrives at the correct final answer, demonstrating the synergy between perception and reasoning.

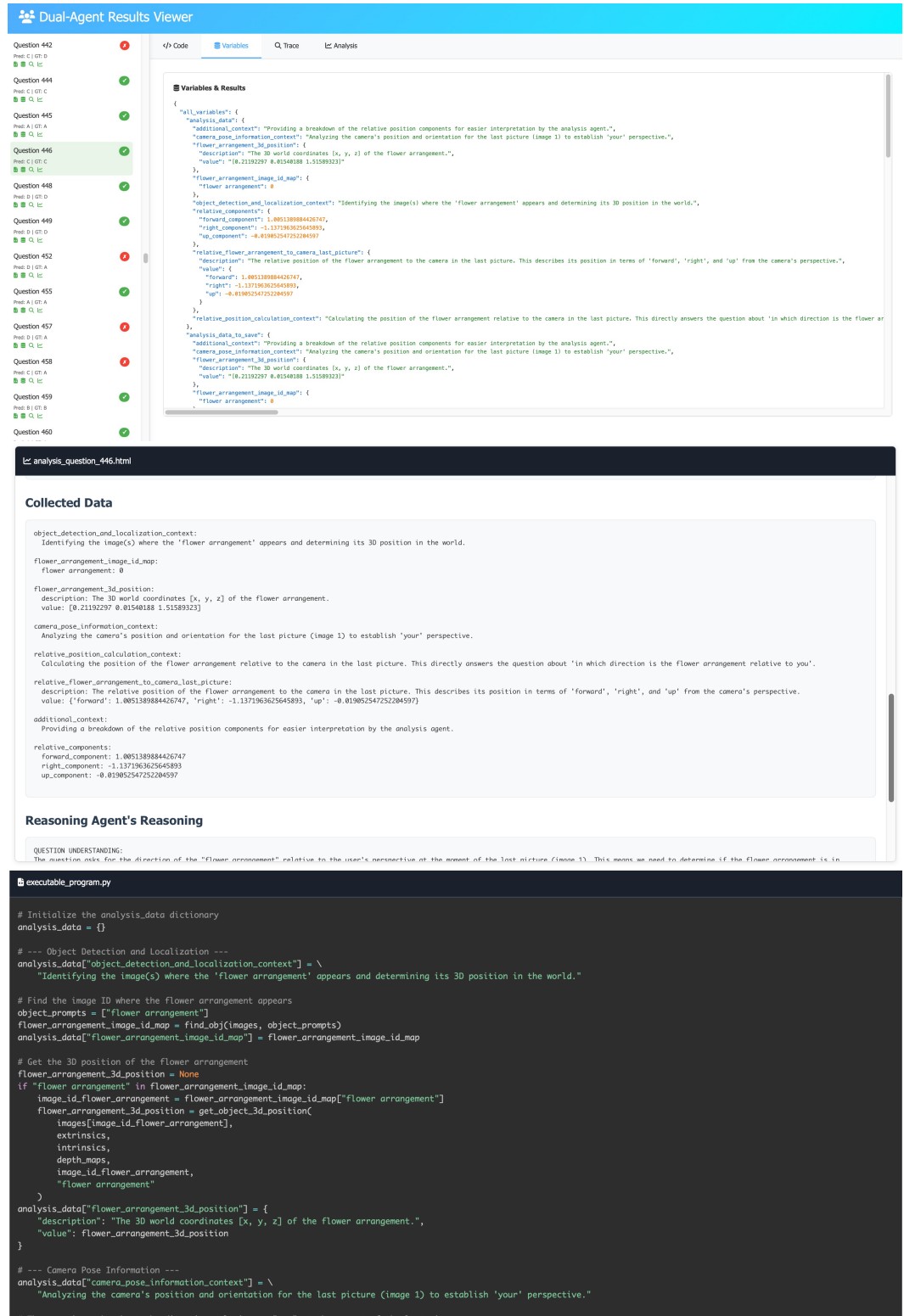

Figure 17: Web-based visualization tool for MSSR. The interface integrates code, execution traces, and reasoning trajectories in a unified view, enabling convenient inspection of the full problem-solving process.

