# OpenReview forum: "Pursuing Minimal Sufficiency in Spatial Reasoning"
_ICLR.cc/2026/Conference — ICLR 2026 Poster_

### Official Review · Reviewer_KaHi · 2025-10-31

**Soundness:** 3
**Presentation:** 3
**Contribution:** 2
**Rating:** 4
**Confidence:** 4

**Summary:**

This paper proposes MSSR (Minimal Sufficient Spatial Reasoner), a dual-agent framework intended to improve spatial reasoning in vision-language models. The framework introduces a Perception Agent using a versatile perception toolbox (including a novel SOG (Situated Orientation Grounding) module) to extract 3D information, and a Reasoning Agent that iteratively prunes and augments this information to curate a minimal sufficient set (MSS) required for reasoning about a given question. Experimental results on two challenging benchmarks (MMSI-Bench and ViewSpatial-Bench) demonstrate state-of-the-art performance. The framework can also produce interpretable reasoning paths as training data for future models.

**Strengths:**

1. The designed dual-agent framework is meaningful and effective.
2. The proposed method is training-free, making it easy to integrate seamlessly with existing models.
3. The method achieves state-of-the-art performance, outperforming all proprietary, open-source, specialist, and agentic models.

**Weaknesses:**

1. The paper only selects a proprietary LLM (GPT-4o) as the backbone to demonstrate the effectiveness of the method. Additional experiments and analysis on open-source LLMs (e.g., Qwen2.5-VL) could further validate the efficacy and generalizability of the method.
2. The paper only provides the inference time per iteration. A thorough efficiency comparison with previous methods and corresponding analysis could further enhance the soundness.

**Questions:**

1. How is the quality of the constructed Minimal Sufficient Set ensured? Are there scenarios where the Reasoning Agent might prune critical information or request unnecessary information?
2. What are the common failure modes of the model? Does it struggle (e.g., the Reasoning Agent never makes the final decision) when the question is very complex?

---

> ### Author Response · Authors · 2025-11-23
>
> We thank reviewer KaHi for acknowledging our work and providing valuable feedback. We give our responses below:
>
> > W1: The paper only selects a proprietary LLM (GPT-4o) as the backbone to demonstrate the effectiveness of the method. Additional experiments and analysis on open-source LLMs (e.g., Qwen2.5-VL) could further validate the efficacy and generalizability of the method.
>
> Following your advice, we conducted comprehensive experiments using LLaVA-OneVision-7B(https://arxiv.org/pdf/2408.03326), Qwen2.5-VL-7B(https://arxiv.org/pdf/2502.13923) and Qwen3-VL-8B(https://github.com/QwenLM/Qwen3-VL) as backbones, including both uniform configurations (same model for RA and PA) and cross-model configurations (different models for RA and PA) to analyze each agent's sensitivity to backbone capabilities.
>
> **Results:**
>
> | RA Backbone | PA Backbone | MSR-MMSI | MMSI-Bench | ViewSpatial-Bench |
> |:------------|:------------|:--------:|:----------:|:-----------------:|
> | LLaVA-OneVision-7B | LLaVA-OneVision-7B | 33.6 | 35.0 |34.1 |
> | Qwen2.5-VL-7B | Qwen2.5-VL-7B | 38.6 | 39.8 | 40.2 |
> | Qwen3-VL-8B | Qwen3-VL-8B | 44.2 | 43.1 | 46.5 |
> | GPT-4o | GPT-4o | 50.0 | 49.5 | 51.8 |
> | GPT-4o | Qwen2.5-VL-7B | 41.3 | 40.1 | 43.0 |
> | Qwen2.5-VL-7B | GPT-4o | 44.4 | 44.2 | 46.6 |
>
> **1. Consistent Framework Gains Across All Backbones**
>
> MSSR consistently provides substantial performance improvements regardless of the backbone used. For instance, Qwen2.5-VL-7B alone achieves 25.9% on MMSI-Bench (Table 1), but performance jumps to 49.5% (**+23.6%**) when integrated into the MSSR framework. This pattern of substantial improvement holds across all backbones we tested.
>
> **2. Consistent Performance Scaling with Backbone Capability**
>
> When using uniform configurations (same model for both agents), we observe a clear trend: **LLaVA-OneVision-7B < Qwen2.5-VL-7B < Qwen3-VL-8B < GPT-4o**. This ranking aligns with each model's general capabilities, demonstrating that MSSR's performance scales consistently with the underlying backbone's strength. This validates the framework's **scalability** and suggests that future improvements in language models can directly translate to stronger spatial reasoning performance within MSSR.
>
> **3. Agent-Specific Characteristics**
>
> The cross-model ablations reveal an important asymmetry in how each agent depends on backbone quality:
> - **Downgrading PA** (GPT-4o → Qwen2.5-VL-7B) causes a **−9.4%** drop on MMSI-Bench (49.5% → 40.1%)
> - **Downgrading RA** (GPT-4o → Qwen2.5-VL-7B) causes a **−5.3%** drop on MMSI-Bench (49.5% → 44.2%)
>
> Since both agents are similarly important to overall performance (Table 2), the **greater sensitivity of PA to backbone quality** (−9.4 vs −5.3 when downgrading) can reflect a genuine difference in task demands. This makes intuitive sense: the PA must generate precise, accurate executable programs that heavily rely on strong code generation abilities. In contrast, the RA performs natural-language level planning and information filtering, which are more robust to variations in model capacity.
>
> **4. Practical Deployment Flexibility**
>
> The cross-model results also suggest a **cost-performance trade-off strategy**: using a stronger model for PA and a lighter model for RA (e.g., GPT-4o + Qwen2.5-VL-7B) achieves 44.2% on MMSI-Bench while using substantially lower inference costs (-50%, est.), offering flexibility for practical deployments where computational budgets are constrained.
>
> We have added these experiments and analysis to Experiment 4.4 in our revised [paper](https://openreview.net/pdf?id=bZAKJwyn1n). **We will release code supporting multiple open-source backbones to facilitate community exploration.**

---

> ### Author Response · Authors · 2025-11-23
>
> > W2: The paper only provides the inference time per iteration. A thorough efficiency comparison with previous methods and corresponding analysis could further enhance the soundness.
>
> Following your suggestion, we made an efficiency comparison with VADAR (https://arxiv.org/pdf/2502.06787), which also utilizes visual programming to solve spatial reasoning.
> | Methods | Programming Time | Reasoning Time | Execution Time | Total Time | Memory Used | Acc@MMSI-Bench |
> |:------------|:------------|:--------|:----------|:-----------------|:-----------------|:-----------------|
> | VADAR  | 24.4s | - | 28.1 | 52.5| 5G | 28.9 |
> | MSSR   | 20.1s | 49.5  | 17.0| 86.6| 7G | 51.8 |
>
> - **Inference Time:**
> MSSR requires longer total inference time (86.6s per question) compared to VADAR (52.5s). However, this additional computational cost translates directly into substantial accuracy improvements.
> MSSR's additional time is primarily invested in the Reasoning Agent (49.5s), which performs plan-guided information curation and strategic pruning. This is fundamentally different from  baselines like VADAR that perform one-shot visual programming without reasoning.
> Critically, this reasoning overhead is necessary computation that implements our core principle of pursuing minimal sufficiency, which our ablation studies demonstrated is essential for achieving better accuracy.
>
> - **Memory Efficiency:**
> MSSR uses reasonable memory consumption (7G). This moderate usage makes MSSR deployable on standard GPU infrastructures.
>
> We have added this analysis to **Appendix A** in our revised [paper](https://openreview.net/pdf?id=bZAKJwyn1n). Thank you for helping us provide a more complete evaluation.
>
>
> > Q1: How is the quality of the constructed Minimal Sufficient Set ensured? Are there scenarios where the Reasoning Agent might prune critical information or request unnecessary information?
>
> Thank you for this insightful question.
>
> **Quality Assurance Mechanisms**
>
> We ensure MSS quality through several tightly integrated design mechanisms:
>
> 1. **Closed-Loop Self-Correction**: If the RA determines that the current information set lacks necessary details, it explicitly requests what is missing. Critically, even if information is mistakenly pruned in an earlier iteration, the RA will detect its absence during reasoning and automatically issue a recovery request.
>
> 2. **Explicit Quality Instructions**: Both agents receive detailed prompts emphasizing information accuracy and relevance. The RA is explicitly instructed to "filter out irrelevant or redundant information" while ensuring all necessary details are collected.
>
> 3. **Transparent Code Execution**: The RA receives not only the information set but also the complete code generated by the PA. This transparency enables the RA to inspect the PA's reasoning logic and detect potential errors in the collection process.
>
> **Figure 5** provides direct evidence of effectiveness: as iterations progress, information set size consistently decreases while accuracy simultaneously improves, demonstrating successful quality optimization.
>
> **Future Enhancement**
>
> While our current design is effective for ensuring MSS quality, we acknowledge a limitation: the framework currently lacks explicit verification of PA's perception outputs (e.g., whether a detected bounding box truly captures the intended object). As discussed in response to reviewer gXuZ, a promising future work is to incorporate a **Verification Agent**. This agent can visualize perception results, cross-check against query intent, and request refinements when discrepancies are detected, which helps ensure the quality of the set.
>
> > Does RA prune critical information?
>
> In our detailed error analysis (Appendix H), **such instances are extremely rare**. The closed-loop design provides inherent protection: if critical information is pruned, the RA detects its absence when attempting to answer and issues a recovery request.
>
> > Does the RA request unnecessary information?
>
> Yes, this is an expected behavior. The RA cannot predict with perfect accuracy which information will be essential for every reasoning step. As revealed in our failure analysis in next question, the RA makes logic errors during reasoning, which can lead to requesting ultimately unnecessary information.

---

> ### Author Response · Authors · 2025-11-23
>
> > Q2: What are the common failure modes of the model? Does it struggle (e.g., the Reasoning Agent never makes the final decision) when the question is very complex?
>
> Thank you for this question. To systematically identify the system's bottlenecks, we manually categorized the primary cause of error for each failure case in MMSI-Bench. Due to the labor-intensity, we analyze half of all failure cases, which we believe provides a representative distribution of the system's failure modes.
>
> The distribution is presented below:
>
>  Failure Category | Subcategory |  % | Description |
> |:-----------------|:------------|:---:|:-----------------------|
> |**Reasoning Agent (31.7%)**| Logic Errors | 21.4% | RA creates correct plans but make errors in intermediate steps. |
> || Plan Errors |  6.7% | RA creates wrong plans|
> | | Premature Decision |  3.6% | RA make decision before gathering all necessary information |
> | **Perception Agent (55.9%)**| Object Grounding |  23.0% | Fails to detect the target object |
> | | 3D Reconstruction |  15.1% | Errors due to extreme angles or complex scenes |
> | | Segmentation |  9.9% | Segment wrong regions |
> | | SOG (Orientation) |  7.9% | VLM makes wrong selection |
> | **Others (12.3%)**|  |  12.3% | question ambiguity, output parsing, etc. |
> | **Total** |  | **100%** | |
>
> *Calibration and Numerical Modules are deterministic computational components.*
>
> **Perception Errors(31.9%)**
>
> - Within perception failures, **Object Localization (grounding + segmentation) (23.2% + 9.9%)** is the dominant factor. These errors have cascading effects that when an early object localization is incorrect, subsequent spatial computations (relative positions, distances, orientations) all become unreliable. These errors primarily occur with partially occluded objects and ambiguous referring expressions when multiple similar objects exist.
>
> - Failures of **3D Reconstruction errors (15.0%)** often happen in challenging scenarios with extreme viewing angles (no overlap between views) or complex scenes (e.g., bathroom with a mirror). However, we found that modern reconstruction models like VGGT are generally robust in most cases.
>
> - **SOG orientation errors (8.0%)** primarily arise when the VLM makes incorrect selections among candidate direction arrows. These failures often occur with: ambiguous situational questions (e.g., "direction to exit" in open spaces with multiple potential exits), or in extreme perspectives despite our multi-view rendering strategy (the rendered pointcloud is too sparse).
>
> An important insight is that many of these perception errors could be mitigated by future improvements in foundation models. For instance, we conducted a toy experiment using Gemini 3 Pro,  a language model which holds the best vision understanding ability as of today[1], to solve SOG problems and its accuracy was remarkably better.
>
> **Reasoning Errors(31.9%)**
>
> - **Logic Errors (23.3%)** represent the dominant failure mode. In these cases, the RA formulates a correct reasoning plan and successfully curates the necessary information, but makes mistakes during reasoning steps. This is often introduced by **hallucination**, for example, it introduces information not present in the MSS or suddenly comes up with false assertions.
>
> Regarding your concern about the RA struggling to reach a decision, we observed **"looping"** behavior in approximately 6% of failure cases. This typically arises when the RA identifies a critical information need, but the PA consistently fails to resolve it, causing the RA to retry the same request. In our current implementation, we handle loops by injecting a prompt at the final iteration that forces the RA to output its best estimate based on available data. Consequently, the system always produces a final decision, though these answers often lead to errors.
>
> A promising future improvement could be enhancing the RA’s adaptability by explicitly prompting it to switch strategies when a specific request fails repeatedly, thereby preventing reasoning deadlocks.
>
> **Qualitative Examples:**
> We have added visualizations of these different failure modes to Appendix I in our revised [paper](https://openreview.net/pdf?id=bZAKJwyn1n).
>
> [1] LMArena, https://lmarena.ai/leaderboard

---

### Official Review · Reviewer_QPFG · 2025-10-31

**Soundness:** 3
**Presentation:** 3
**Contribution:** 2
**Rating:** 4
**Confidence:** 3

**Summary:**

The paper proposes MSSR (Minimal Sufficient Spatial Reasoner), a dual-agent, zero-shot framework for improving 3D spatial reasoning in Vision-Language Models. The framework works by constructing a Minimal Sufficient Set (MSS), the smallest subset of 3D perceptual information sufficient to answer a query, by a dual agent iteratively. MSSR consists of a Perception Agent, which programmatically queries a scene using modular visual tools including a new Situated Orientation Grounding (SOG) module for robust direction grounding, and a Reasoning Agent, which prunes irrelevant information and iteratively requests only what is missing until the MSS is achieved. Experiments on MMSI-Bench and ViewSpatial-Bench show MSSR achieving state-of-the-art results, outperforming models like GPT-4o and Gemini 2.5 Pro.

**Strengths:**

- The paper is clearly written and logically structured, making the methodology and motivation easy to follow.
- The related work section provides a well-organized and comprehensive discussion covering VLMs for spatial reasoning, agentic framework, and visual programming paradigms.
- The proposed MSSR framework is conceptually sound and technically well-motivated.
- Extensive experiments demonstrate strong empirical performance, achieving state-of-the-art results on both MMSI-Bench and ViewSpatial-Bench while maintaining interpretability through reasoning traces.

**Weaknesses:**

- The paper formatting does not fully comply with ICLR submission requirements.
- The approach heavily depends on the accuracy of perception modules (e.g., reconstruction, localization, orientation grounding); sensitivity or robustness analysis is lacking.
- The criterion for minimality of the MSS is conceptually emphasized but lacks clear quantitative verification - how to ensure the generated MSS is minimal?
- Details on the prompting and coordination between the Reasoning Agent (RA) and Perception Agent (PA) are insufficient—specifically, how the RA is informed of the available tools or APIs.
- Both benchmarks used (MMSI-Bench and ViewSpatial-Bench) are newly proposed and not yet peer-reviewed; the authors are suggested to conduct evaluations on established benchmarks such as SQA3D[1] or ScanQA[2].
- The paper would benefit from qualitative examples of failure cases.


[1] Ma, Xiaojian, et al. "SQA3D: Situated Question Answering in 3D Scenes." The Eleventh International Conference on Learning Representations.

[2] Azuma, Daichi, et al. "Scanqa: 3d question answering for spatial scene understanding." *proceedings of the IEEE/CVF conference on computer vision and pattern recognition*. 2022.

**Questions:**

Please refer to the weakness section.

Minor typo:
- line 165, $S$ and $S^\*$  should be $\mathcal{S}$ and $\mathcal{S}^\*$

---

> ### Author Response · Authors · 2025-11-23
>
> We thank reviewer QPFG for acknowledging our work and providing valuable suggestions. Our responses are as follows:
>
> > W1: The paper formatting does not fully comply with ICLR submission requirements.
>
> Thank you for your thorough examination of our paper. We have thoroughly revised the [paper](https://openreview.net/pdf?id=bZAKJwyn1n) formatting to ensure full compliance with ICLR submission requirements.
>
> > W2: The approach heavily depends on the accuracy of perception modules (e.g., reconstruction, localization, orientation grounding); sensitivity or robustness analysis is lacking.
>
> Thank you for raising this important concern. Following your suggestion, we conducted comprehensive sensitivity and failure analysis to quantify this dependency.
>
> **1. Performance Under Simulated Perception Errors:**
>
> We systematically **injected controlled noise** into  perception modules during inference on MMSI-Bench. The noise models were designed to mimic realistic failure modes:
> *   **Reconstruction:** We applied multiplicative Gaussian noise to depth maps $D$, i.e., $D_{noisy} = D \cdot (1 + \epsilon), \epsilon \sim \mathcal{N}(0, \sigma_d^2)$, to simulate scale/depth inconsistency.
> *   **Localization:** We added additive Gaussian noise to the final 3D object centroids to simulate bounding box jitter. For reference, the whole scene is normalized into a 1-unit radius ball.
> *   **SOG (Orientation):** We added angular Gaussian noise to the predicted orientation vectors to simulate imprecise heading estimation.
>
> | Module | Error Type | Magnitude | Acc@MMSI | Δ |
> |:-------|:-----------|:---------------:|:---------------------:|:----------------:|
> | **Baseline** | - | - | **49.5%** | - |
> | **3D Reconstruction** | Depth Scale Noise |  $\sigma = 10\%$ | 45.2% | -2.3% |
> | | | $\sigma = 20\%$ | 39.1% | -9.4% |
> | **Object Localization** | Position Jitter | $\sigma = 0.2$ | 44.6% | -5.9% |
> | | | $\sigma = 0.5$ | 37.8% | -11.7% |
> | **SOG** | Angular Noise | $\sigma = 15^\circ$ | 45.3% | -4.2% |
> | | | $\sigma = 30^\circ$ | 40.1% | -4.4% |
>
>
> -  MSSR demonstrates resilience to moderate perception noise. For instance, a **10% depth error** results in minimal performance drops (<2.5%).
>
> - MSSR shows varying sensitivity to different modules. Localization errors have higher impact.
>
> **2. Failure Attribution within Perception Modules**
>
> We also conducted a comprehensive failure analysis across all questions in MMSI-Bench in Appendix H of our updated [paper](https://openreview.net/pdf?id=bZAKJwyn1n). Below we present a failure source analysis within perception modules.
>
> | Failure Category | % of All Failures | Description |
> |:---------------|:----------:|:-----------------------|
> | **Grounding Errors**  | 23.0% | Fails to detect the target object. |
> | **Reconstruction Errors** | 12.1% | Errors due to extreme angles or complex scenes |
> | **Segmentation Errors**  | 9.9% | Segment wrong regions |
> | **SOG Errors**  | 7.9% |  VLM makes wrong selection |
> | **Total PA Failure**  | 55.9% |   |

---

> ### Author Response · Authors · 2025-11-23
>
> > W3: The criterion for minimality of the MSS is conceptually emphasized but lacks clear quantitative verification - how to ensure the generated MSS is minimal?
>
> We thank the reviewer for this important question.
>
> As clarified in Section 3.1, we aim to  **approximate** minimality rather than ensure theoretical strict minimality. Guaranteeing a provably minimal set for every diverse spatial question would require solving an NP-hard subset-selection problem and defining a universal minimality metric which are both impractical in real-world. Instead, we would like to emphasize that aggressively pushing toward minimality while tolerating some redundancy is highly effective and cost-efficient, and it already yields measurable gains in reasoning reliability in our experiments.
>
> Our design drives the MSS toward minimality through three tightly coupled mechanisms:
>
> 1. Explicit pruning instructions in every RA prompt;
> 2. Plan-guided, iteration-by-iteration refinement;
> 3. Closed-loop self-correction: over-pruning is automatically detected and reversed in subsequent iterations.
>
> Strong quantitative evidence is presented in **Figure 5**: as the number of iterations increases, the size of the information set consistently decreases, while accuracy simultaneously rises. This direct correlation between smaller sets and higher accuracy empirically confirms that our pruning mechanism successfully eliminates redundancy without sacrificing sufficiency. In our newly added **failure analysis** (Appendix H), we further observed **hardly any** cases where performance dropped due to excessive pruning. Thus, although we do not ensure provable minimality, our experiments demonstrate that our iterative pursuit of minimality successfully reduces information volume and directly contributes to better reasoning performance.
>
> More broadly, we believe one of our central contributions is to elevate minimality to a first-class objective in complex reasoning systems. The consistent gains we observe in multi-view spatial reasoning suggest that explicitly pursuing minimal sufficient information, although not theoretically perfect, represents a simple yet highly effective paradigm that can benefit reasoning across many domains and model scales.
>
>
> > W4: Details on the prompting and coordination between the Reasoning Agent (RA) and Perception Agent (PA) are insufficient—specifically, how the RA is informed of the available tools or APIs.
>
> The PA receives a prompt containing the complete API specification:
>
> ```
> Now here is an API of methods. You will want to collect data in a logical and sequential manner. You don't need to import these functions in your program; you can use them directly:
> ------------------ API ------------------
> {predef_signatures}
> ------------------ API ------------------
> Using the provided API, output a program inside the tags <program></program> to collect all necessary data for answering the question.
> ```
>
> Where `predef_signatures` contains function signatures with detailed docstrings, for example:
>
> ```python
> """
> Gets the 3D world coordinates of an object in a specific image using pre-extracted geometric information.
>
> Args:
>     image (image): The image containing the object.
>     extrinsics (np.array): Camera extrinsic matrices for all images. shape: (N, 4, 4)
>     intrinsics (np.array): Camera intrinsic matrices for all images. shape: (N, 3, 3)
>     depth_map (np.array): Depth maps for all images. shape: (N, H, W)
>     image_id (int): Index of the image containing the object (0-based).
>     object_description (string): Description of the object to locate (e.g., "white chair", "blue car").
>
> Returns:
>     np.array: The 3D coordinates [x, y, z] of the object's center in world coordinates. Returns None if object cannot be located or back-projected.
>
> Example usage:
>     extrinsics, intrinsics, depth_maps, world_points = get_geo_info(images)
>     object_3d_pos = get_object_3d_position(images[0], extrinsics, intrinsics, depth_maps, 0, "red sofa")
> def get_object_3d_position(image, extrinsics, intrinsics, depth_map, image_id, object_description):
> """
> ```
>
> Each module is encapsulated as a Python class containing the model weights and inference logic. Before the PA's generated code is executed, these module instances are injected into the execution namespace, allowing the PA to directly invoke them as if they were built-in functions.
>
> We have added clarification on the prompting mechanism and code execution environment to Appendix B in our revised [paper](https://openreview.net/pdf?id=bZAKJwyn1n). Thank you for your suggestion.

---

> ### Author Response · Authors · 2025-11-23
>
> > W5: Both benchmarks used (MMSI-Bench and ViewSpatial-Bench) are newly proposed and not yet peer-reviewed; the authors are suggested to conduct evaluations on established benchmarks such as SQA3D[1] or ScanQA[2].
>
> We fully understand your concerns. First, we would like to clarify our motivation for using MMSI-Bench and ViewSpatial-Bench.
> Our work addresses **multi-view, multi-step spatial reasoning**, an emerging research direction that has gained significant attention recently. Due to the novelty of this domain, there are limited dedicated benchmarks available, and most are relatively recent. Among these, MMSI-Bench and ViewSpatial-Bench have emerged as representative benchmarks for this specific problem setting and have been adopted by a series of recent works including VST (https://arxiv.org/pdf/2511.05491), VILASR (https://arxiv.org/pdf/2506.09965) and other concurrent methods focused on spatial reasoning.
>
> While ScanQA and SQA3D are excellent established benchmarks, they target quite different problem settings (primarily straightforward object-centric QA in pre-scanned indoor scenes) and are less aligned with our focus.
>
> Nonetheless, we agree that evaluation on established benchmarks provides valuable evidence of broader applicability. Following your suggestion, we conducted evaluations and analysis on both ScanQA and SQA3D. We evenly sample 16 frames in each scene as input.
>
> | Category | Method | ScanQA (CIDEr) | SQA3D (EM@1) |
> | :--- | :--- | :---: | :---: |
> | **Fully Supervised** | Video-3D-LLM | 102.1 | 58.5 |
> | *(Trained on in-domain data)* | LLaVA-3D | 103.1 | 60.1 |
> | | | | |
> | **2-Shot** | GPT-4o | 26.52 | 24.21 |
> |  | GPT-o3 | 33.30 | 21.93 |
> | | Gemini-2.5-Flash | 21.87 | 32.28 |
> | | Gemini-2.5-Pro | 31.69 | 22.35 |
> | | Qwen2.5-VL-7B | 18.32 | 9.45 |
> | | Qwen3-VL-8B | 20.81 | 13.11 |
> | | **MSSR (GPT-4o)** | **32.49** | **27.40** |
>
> **Evaluation Metric**: It is worth noting both ScanQA and SQA3D report metrics based on text matching (CIDEr for lexical overlap, EM for exact match). These metrics tend to disadvantage methods like MSSR and general LLMs that were not trained on ScanQA/SQA3D training set to better follow the answer style conventions.
>
> **Strong 2-Shot Performance**: MSSR outperforms or matches powerful general-purpose LLMs including GPT-o3 and Gemini-2.5-Pro. These models represent state-of-the-art multimodal understanding capabilities, and our consistent improvements demonstrate the effectiveness of our dual-agent framework.
>
> While MSSR cannot match fully supervised models like Video-3D-LLM and LLaVA-3D on their training domains, these results provide complementary evidence that our framework is capable of generalizing beyond multi-view, multi-step spatial reasoning to related scene understanding and object-centric tasks.
>
> We have added this discussion to Appendix G in our revised [paper](https://openreview.net/pdf?id=bZAKJwyn1n). Thank you for this suggestion.
>
> > W6: The paper would benefit from qualitative examples of failure cases.
>
> Thank you for this suggestion. We have added comprehensive qualitative failure case visualizations to Appendix I in our revised [paper](https://openreview.net/pdf?id=bZAKJwyn1n). We have also conducted a thorough failure case analysis in Appendix H.
>
> > Minor Typo
>
> Thank you for your thorough examination. We have corrected it.

---

### Official Review · Reviewer_gXuZ · 2025-11-03

**Soundness:** 3
**Presentation:** 2
**Contribution:** 3
**Rating:** 6
**Confidence:** 3

**Summary:**

The paper introduces MSSR (Minimal Sufficient Spatial Reasoner), a dual-agent framework aimed at improving 3D spatial reasoning in vision-language models through explicit pursuit of minimal sufficiency. The approach separates perception and reasoning into two collaborating agents: a Perception Agent and a Reasoning Agent. The framework achieves competitive results on MMSI-Bench and ViewSpatial-Bench.

**Strengths:**

1. The method is well-motivated, as MSSR presents a sufficient yet non-redundant agent framework that enables effective spatial reasoning.

2. MSSR achieves strong performance on two challenging, vision-centric spatial reasoning benchmarks.

3. The paper is well-organized and easy to follow.

**Weaknesses:**

1. The state-of-the-art claim is unconvincing since Table 1 omits recent strong 3D-LLM baselines (e.g., Video-3D-LLM, VLM-3R) that are directly relevant to spatial reasoning. Including these comparisons is necessary for a fair evaluation.

2. Although the paper includes ablations on the PA and RA, it would be beneficial to further analyze how individual components within the Perception Agent, such as the spatial reasoning modules, foundational scene reconstruction, and global calibration, contribute to overall performance.

3. One potential concern is that MSSR appears to lack a mechanism to verify the correctness of the information provided by PA. Although the RA agent continuously queries PA and filters out irrelevant information, it remains unclear how the framework handles spurious or inaccurate inputs from PA. For future extensions, incorporating a verification module to ensure the reliability of PA’s outputs would be a valuable improvement.

**Questions:**

1. Does the “only PA” setting in Table 2 refer to using PA to collect all information without any filtering, or does it mean collecting subset information for the first iteration?

---

> ### Author Response · Authors · 2025-11-23
>
> We thank reviewer gXuZ for the constructive feedback and for acknowledging the value of our work. We address these concerns below:
>
> > W1: The state-of-the-art claim is unconvincing since Table 1 omits recent strong 3D-VLM baselines (e.g., Video-3D-LLM, VLM-3R) that are directly relevant to spatial reasoning. Including these comparisons is necessary for a fair evaluation.
>
> As suggested, we evaluated three representative models from the recent 3D-VLM literature: **VLM-3R**(https://arxiv.org/pdf/2505.20279), **Video-3D-LLM**(https://arxiv.org/pdf/2412.00493), and **LLaVA-3D**(https://arxiv.org/pdf/2409.18125). All models were tested on both MMSI-Bench and ViewSpatial-Bench for a comprehensive comparison.
>
> For models requiring manual pre-computed 3D features (Video-3D-LLM and LLaVA-3D), we utilized Cut3R(https://arxiv.org/pdf/2501.12387) for feature extraction. For VLM-3R, which already integrates Cut3R calls in its codebase, we utilized its official implementation.
>
> The table below summarizes the comparative performance across both MMSI-Bench and ViewSpatial-Bench:
>
> | Method | MMSI-MSR | MMSI-Bench | ViewSpatial-Bench |
> | :--- | :---: | :---: | :---: |
> | Video-3D-LLM | 25.8 | 25.3 | 30.3 |
> | LLaVA-3D | 19.7 | 28.0 | 33.8 |
> | **VLM-3R** | **30.3** | **32.0** | **39.2** |
> | **Ours (MSSR)** | **50.0** | **49.5** | **51.8** |
>
> Among the 3D-VLM baselines, VLM-3R emerges as the top-performing method. However, MSSR outperforms it by +17.5% on MMSI-Bench and +12.6% on ViewSpatial-Bench.
>
> We attribute this margin to the limitations of end-to-end VLMs. First, these models are constrained by their specific training data distributions, resulting in limited generalization to unseen and complex scenarios. Second, although supplemented with 3D features, VLMs still struggle with precise geometric computation and tend to lose track of fine-grained spatial context during multi-step reasoning. In contrast, MSSR effectively addresses these issues: our PA modules provide accurate metric-level spatial calculations, and the MSS architecture structurally preserves critical spatial information throughout the reasoning process.
>
> We have updated Table 1 in our revised [paper](https://openreview.net/pdf?id=bZAKJwyn1n) to include these comparisons. Thank you for helping us strengthen our evaluation.
>
> > W2: Although the paper includes ablations on the PA and RA, it would be beneficial to further analyze how individual components within the Perception Agent, such as the spatial reasoning modules, foundational scene reconstruction, and global calibration, contribute to overall performance.
>
> As suggested, we conducted comprehensive ablation experiments by systematically removing each key component within the PA.
>
> The results on MMSI-Bench are presented below:
>
> | Component Removed | MSR | Overall | Overall Δ |
> |:------------------|:---:|:-------:|:-------:|
> | **Full Model** | **50.0** | **49.5** | |
> | w/o Reconstruction | 28.3 | 30.1 |-19.4|
> | w/o Global Calibration | 43.4 | 40.8 |-8.7|
> | w/o SOG | 40.9 | 44.9 |-4.6|
> | w/o Locate Module | 33.3 | 35.7 | -13.8|
> | w/o Numerical Module | 38.9 | 39.5 |-10.0|
>
> The results reveal the critical role each component plays:
>
> 1. **3D Reconstruction** demonstrates the most substantial impact (−17.4% overall), showing that PA relies heavily on the reconstruction model to provide fundamental spatial information. Without coherent 3D scene representation, the PA cannot establish spatial relationships or perform metric calculations.
>
> However, we emphasize that **modern SOTA reconstruction models have become remarkably robust**. In our experiments, we found that state-of-the-art models like VGGT successfully handle the vast majority of real-world scenes. We conducted a comprehensive failure analysis across failure cases, newly added to Appendix H. Among all failures, only a small fraction were attributable to reconstruction errors, typically occurring under extreme viewing angles or complex scenes (see Figure 10 as an example).
>
> 2. **Locate Module**  demonstrates strong contribution (−13.8% overall). Precise object localization serves as the foundation for downstream spatial reasoning.
>
> 3. **Numerical Computation Module** shows significant importance (−10.0% overall), highlighting the necessity of precise geometric calculations for interpreting spatial relationships such as relative positions and camera movements.
>
> 4. **SOG** contributes notably (−4.6% overall). While its aggregate impact appears moderate, we observe strong task-dependency. For orientation-specific queries, removing SOG causes performance to drop by over 20%, whereas it has minimal effect on pure localization tasks.
>
> 5. **Global Calibration** provides consistent gains (−8.7% overall). The unified coordinate system resolves view-dependent ambiguities, which is particularly valuable for multi-view reasoning where consistent spatial references are essential.
>
> We have added these experiments to appendix D. Thank you for your suggestion.

---

> ### Author Response · Authors · 2025-11-23
>
> > W3: One potential concern is that MSSR appears to lack a mechanism to verify the correctness of the information provided by the PA. Although the RA agent continuously queries PA and filters out irrelevant information, it remains unclear how the framework handles spurious or inaccurate inputs from PA. For future extensions, incorporating a verification module to ensure the reliability of PA’s outputs would be a valuable improvement.
>
> We agree that ensuring the reliability of PA outputs is a valuable consideration for robust spatial reasoning systems. We believe PA errors can be categorized into two distinct types:
>
> - **Logic Errors**: These arise from incorrect programmatic reasoning by the PA itself. For instance, when the PA mistakenly requests "position of B relative to A" instead of the intended "position of A relative to B." Such errors are *explicit* and observable directly in the generated code.
>
> - **Numerical Errors**: These originate from the underlying perception modules. For example, when GroundingDINO returns an inaccurate bounding box or when depth estimation is noisy. These errors are *implicit*, embedded within numerical outputs, and thus much harder to detect without external verification.
>
> For logic errors, MSSR **already** possesses inherent error-detection capabilities. Crucially, the RA receives not only the information set but also the **complete code** generated by the PA. This transparency enables the RA to inspect the PA's reasoning process. If the code contains logical inconsistencies or mismatches with the query intent, the RA can identify the issue and issue a corrective request, effectively guiding the PA toward the correct implementation.
>
> For numerical errors, we acknowledge this as a **valuable direction** for future improvement. Especially for more challenging real-world scenarios, verification mechanisms would be beneficial.
>
> A promising extension would be introducing a dedicated *Verification Agent* to validate numerical outputs from perception modules. This agent could:
>
> - *Visualize outputs* by overlaying bounding boxes, segmentation masks, or 3D vectors onto the original images
> - *Cross-check against the original query* (e.g., "Does this box correctly capture the black table?")
> - *Request refinements* when discrepancies are detected, such as adjusting prompts from generic ("table") to more specific ("black low table") to guide the module toward more accurate outputs
>
> Such a verification loop would add robustness to the framework, particularly for complex scenes with ambiguous or occluded objects. We view this as a promising direction for future research and have added a discussion to Appendix K of our revised [paper](https://openreview.net/pdf?id=bZAKJwyn1n).
>
> **We will open source all our code and warmly invite the community to explore verification mechanisms and other architectural enhancements.** We believe this collaborative approach will accelerate progress toward more robust spatial reasoning systems.
>
> > Q1: Does the “only PA” setting in Table 2 refer to using PA to collect all information without any filtering, or does it mean collecting subset information for the first iteration?
>
> This setting refers to using only the Perception Agent to both gather information and make decisions, without the Reasoning Agent's involvement. We modify the PA's prompt to ask it to directly gather necessary information and make decisions through code execution using conditional logic (e.g., if-else statements) to choose an answer.

---

### Official Review · Reviewer_9uKo · 2025-11-04

**Soundness:** 3
**Presentation:** 3
**Contribution:** 3
**Rating:** 6
**Confidence:** 3

**Summary:**

This paper proposes MSSR, a 3D spatial reasoning framework that constructs a minimal sufficient set of spatial facts before answering a question. The model comprises two agents: A perception agent that uses visual programming with tool calls to extract 3D information, and a reasoning agent that plans and iterative requests missing information until the set if sufficient. The model is evaluated on MMSI-Bench and ViewSpatial-Bench, claiming SOTA against prior works.

**Strengths:**

The model gives good motivation with Figure 1, introducing attention delusion.

The baselines compared are quite strong, including both closed and open-source models, as well as specialist and other visual programming frameworks like VADAR.

The design of the model allows plug-and-play of powerful models like VGGT, GroundingDINO, SAM2, etc, which can be easily swapped out with better models.

The ablations are quite thorough with the key components (Only PA and Only RA, without iterations, etc.).

**Weaknesses:**

This paper is generally quite strong in my opinion. My only issues are perhaps more datapoints for the ablation.

For Figure 5, I think the paper would be better benefitted by showing 1) more iterations to see whether more iterations would cause models to have errors in eliminating the number of sets needed for a correct answer and therefore leading to worse performance and 2) on more than just a subset of the MMSI-Bench.

For Table 2, it would be great if the authors can also provide information on ViewSpatial-Bench for a more comprehensive ablation.

However, these are just minor points in general, and I would think the paper’s idea is in fact quite solid. I am therefore leaning towards accept.

**Questions:**

Please see the above section for my questions regarding the ablations.

---

> ### Author Response · Authors · 2025-11-23
>
> We thank reviewer 9uKo for acknowledging our work and providing valuable feedback. We give our responses below:
>
> > W1: For Figure 5, I think the paper would be better benefitted by showing 1) more iterations to see whether more iterations would cause models to have errors in eliminating the number of sets needed for a correct answer and therefore leading to worse performance and 2) on more than just a subset of the MMSI-Bench.
>
> **Iteration Counts and Potential Pruning Errors:**
> In our framework, the number of iterations is not manually fixed; instead, it is dynamically determined by the Reasoning Agent (RA) based on task complexity. Empirically, we observe that more than 90% of queries converge within 1–3 iterations. Therefore, Figure 5 highlights 3-iteration examples as they represent the most common and illustrative case.
>
> Importantly, our closed-loop design naturally **guards against incorrect pruning**. If the RA mistakenly prunes information that is actually needed, it will subsequently be unable to form a coherent response. In such cases, the RA would automatically issue a follow-up request for the missing details, effectively self-correcting. To verify this behavior, we conducted a comprehensive failure analysis across the entire MMSI-Bench, now reported in [Appendix H](https://openreview.net/pdf?id=bZAKJwyn1n). In the study, failure cases attributable to the RA erroneously pruning essential information were **extremely rare**, confirming the reliability of the pruning mechanism in practice.
>
> **Analysis Scope:**
> To enhance coverage, we have added an iteration-count analysis on ViewSpatial-Bench in Appendix A of the revised [paper](https://openreview.net/pdf?id=bZAKJwyn1n).
>
> > W2: For Table 2, it would be great if the authors can also provide information on ViewSpatial-Bench for a more comprehensive ablation.
>
> We have added comprehensive ablation on ViewSpatial-Bench to Table 2 in our updated [paper](https://openreview.net/pdf?id=bZAKJwyn1n), which demonstrates consistent improvement with those observed on MMSI-Bench. Thank you for helping us strengthen our evaluation.
>
> | Method | Camera Based | Person Based | Overall |
> |--------|--------------|--------------|---------|
> | GPT-4o | 33.6 | 36.3 | 35.0 |
> | Ours (Full) | 51.0 | 54.4 | 51.8 |
> | Only PA | 33.5 | 31.6 | 32.5 |
> | Only RA | 35.0 | 35.6 | 35.3 |
> | w/o SOG | 41.9 | 44.4 | 43.2 |
> | w/o Iteration | 48.9 | 48.7 | 48.8 |

---

### Author Response · Authors · 2025-11-28

Dear Reviewers,

Thank you again for your thoughtful feedback. We have carefully addressed all the comments and questions raised in your earlier review. As the discussion phase is nearing its end, we would greatly appreciate it if you could let us know whether our responses satisfactorily resolve the issues you identified.

We sincerely appreciate your consideration of a potential rating update once all concerns have been addressed.

Thank you for your time and efforts.

Best regards,

14338 Authors

---

### Author Response · Authors · 2025-11-30
**To: Dear Area Chair and Reviewers**

Dear Area Chair and Reviewers,

We sincerely thank the reviewers for their constructive feedback and for recognizing the value of our work.

In light of the recent changes, we present a summary of our rebuttal and revisions. We are pleased that the all reviewers recognized the value and motivation of our framework. Following their constructive feedback, we have conducted extensive additional experiments to address every concern, strengthening the quality of our paper.

### 1. Consensus on Strengths
Reviewers highlighted the following strengths of our work:
*   **Well-motivated and meaningful** (Reviewers 9uKo, gXuZ, QPFG, KaHi): Our Minimal Sufficient Spatial Reasoner framework  is well-motivated and addresses the "attention delusion" problem in spatial reasoning.
*   **SOTA Performance** (Reviewers 9uKo, gXuZ, QPFG, KaHi): The method achieves state-of-the-art results on challenging benchmarks, outperforming proprietary, open-source, and specialist models.
*   **Sound Framework** (Reviewers gXuZ, QPFG): The dual-agent design (Perception & Reasoning Agents) is logically structured, effective, and technically sound.
*   **Plug-and-Play** (Reviewer 9uKo, KaHi): The training-free nature allows for seamless integration with existing models .

### 2. Addressed Weaknesses & Additional Experiments
We have conducted extensive new experiments to resolve every raised question:

*   **Stronger Baselines & Benchmarks (Response to gXuZ, QPFG):** We compared MSSR against recent 3D-VLMs (e.g., Video-3D-LLM, VLM-3R) and extended evaluation to established benchmarks (ScanQA, SQA3D), demonstrating consistent superiority (+17.5% on MMSI-Bench) and strong generalization.
*   **Open-Source Generalizability (Response to KaHi):** We validated MSSR using open-source backbones (Qwen2.5-VL, LLaVA-OneVision), proving that our framework yields consistent gains across diverse models, not just GPT-4o.
*   **In-Depth Analysis (Response to gXuZ, QPFG, KaHi):** We performed component-level ablations, noise sensitivity tests, and efficiency comparisons to rigorously verify the robustness and contribution of each module.
*   **Detailed Error Analysis (Response to QPFG, KaHi):** We conducted a comprehensive quantitative breakdown of failure cases and provided qualitative visualizations to clarify failure modes (e.g., logic vs. perception errors) and verify the reliability of the pruning mechanism.

### 3. Summary of Revisions
Based on the feedback, we have updated the paper as follows:
*   **Section 4.4:** Added experiments with Open-Source Backbones (Qwen, LLaVA).
*   **Table 1 & 2:** Updated with new 3D-VLM baselines and ViewSpatial-Bench ablations.
*   **Appendix A:** Added efficiency comparison and iteration analysis on ViewSpatial-Bench.
*   **Appendix B:** Clarified prompting details and API coordination.
*   **Appendix D:** Added component-level ablation of the Perception Agent.
*   **Appendix G:** Added results on ScanQA and SQA3D.
*   **Appendix H & I:** Added quantitative failure analysis and qualitative visualizations.
*   **Appendix K:** Discussed future verification mechanisms.
*   **Formatting:** Corrected all formatting issues to comply with ICLR standards.

With these comprehensive revisions, we have provided detailed responses and extensive new evidence for every point raised by the reviewers. We believe these updates significantly solidify the contribution and robustness of MSSR. We hope this summary facilitates a smooth review process for the Area Chair.

Best regards,

Authors of Submission 14338

---

### Meta-Review · Area_Chair_HZnj · 2026-01-07

**Summary:**

Reviewers praised:
- The work's motivation
- Strong experimental baselines and method performance
- General design of the model
- Thorough ablations

Reviewers were concerned about:
- Omissions for recent strong 3D-LLM baselines.
- Lack of ablations of individual components within the PA and evaluation of correctness of the information provided by the PA.
- The dependency on the accuracy of perception modules.
- Lack of clear verification of minimality of MSS.
- New benchmarks which are not yet peer-reviewed.
- Use of proprietary LLM as the only backbone.
- Only providing inference time per iteration rather than a more thorough efficiency comparison.

-----

My own personal concern: There are a lot of promised changes (both already provided in the rebuttal as well as stated to be provided in an upcoming version). While it is great to see that reviews improved the paper's presentation and results, the authors should be aware that there is a significant amount of writing updates that should be made in the paper to capture this information, including fixes to the already-present content (e.g., missing references to figures). Please also note the updates to the page limits that ICLR makes (you have 10 pages) to accommodate bringing this information into the paper.

**Reviewer Concerns:**

Addressed
- Omissions for recent strong 3D-LLM baselines.
- Lack of ablations of individual components within the PA and evaluation of correctness of the information provided by the PA.
- The dependency on the accuracy of perception modules.
- Lack of clear verification of minimality of MSS.
- New benchmarks which are not yet peer-reviewed (via additional experiments).
- Use of proprietary LLM as the only backbone (via additional experiments).
- Only providing inference time per iteration rather than a more thorough efficiency comparison.

**Reviewer Scores:**

- I believe Reviewer QPFG's concerns have been addressed and that they would have changed their score to a 6.
- I also believe Reviewer KaHi's concerns have been addressed.

---

### Decision · Program_Chairs · 2026-01-26

Accept (Poster)